# Dynamics and clinical relevance of maternal mRNA clearance during the oocyte-to-embryo transition in humans

Qian-Qian Sha [1,6], Wei Zheng [2,3,6], Yun-Wen Wu [4], Sen Li [1], Lei Guo [1], Shuoping Zhang [2], Ge Lin [2,5✉], Xiang-Hong Ou [1✉] & Heng-Yu Fan [4✉]

Maternal mRNA clearance is an essential process that occurs during maternal-to-zygotic transition (MZT). However, the dynamics, functional importance, and pathological relevance of maternal mRNA decay in human preimplantation embryos have not yet been analyzed. Here we report the zygotic genome activation (ZGA)-dependent and -independent maternal mRNA clearance processes during human MZT and demonstrate that subgroups of human maternal transcripts are sequentially removed by maternal (M)- and zygotic (Z)-decay pathways before and after ZGA. Key factors regulating M-decay and Z-decay pathways in mouse have similar expression pattern during human MZT, suggesting that YAP1-TEAD4 transcription activators, TUT4/7-mediated mRNA 3′-oligouridylation, and BTG4/CCR4-NOT-induced mRNA deadenylation may also be involved in the regulation of human maternal mRNA stability. Decreased expression of these factors and abnormal accumulation of maternal transcripts are observed in the development-arrested embryos of patients who seek assisted reproduction. Defects of M-decay and Z-decay are detected with high incidence in embryos that are arrested at the zygote and 8-cell stages, respectively. In addition, M-decay is not found to be affected by maternal *TUBB8* mutations, although these mutations cause meiotic cell division defects and zygotic arrest, which indicates that mRNA decay is regulated independent of meiotic spindle assembly. Considering the correlations between maternal mRNA decay defects and early developmental arrest of in vitro fertilized human embryos, M-decay and Z-decay pathway activities may contribute to the developmental potential of human preimplantation embryos.

[1] Fertility Preservation Laboratory, Reproductive Medicine Center, Guangdong Second Provincial General Hospital, 510317 Guangzhou, China. [2] Clinical Research Center for Reproduction and Genetics in Hunan Province, Reproductive and Genetic Hospital of CITIC-XIANGYA, 410008 Changsha, China. [3] College of Life Science, Hunan Normal University, 410006 Changsha, China. [4] Life Sciences Institute, Zhejiang University, 310058 Hangzhou, China. [5] Laboratory of Reproductive and Stem Cell Engineering, Key Laboratory of National Health and Family Planning Commission, Central South University, 410008 Changsha, China. [6] These authors contributed equally: Qian-Qian Sha, Wei Zheng. ✉email: linggf@hotmail.com; ouxh@gd2h.org.cn; hyfan@zju.edu.cn

Maternal-to-zygotic transition (MZT) is an initial step in the early development of all investigated animal species; during MZT, transcripts of maternal genes are removed by degradation and the zygotic genome is activated[1,2]. The exact mechanisms by which the maternal mRNAs are degraded during MZT is a long-standing question in reproductive and developmental biology. Genetic and high-throughput sequencing studies on model systems, including *Drosophila*, zebrafish, and *Xenopus*, have indicated that the elimination of maternal transcripts is accomplished by two sequential pathways: the first pathway is entirely mediated by maternal factors accumulated in the mature oocytes and is thus termed maternal (M)-decay; the second pathway depends on de novo zygotic transcription products after fertilization and is thus termed zygotic (Z)-decay[3–5].

Significant progress has recently been made in understanding the regulation of mRNA stability in mammalian oocytes and zygotes. CNOT6L, which is a catalytic subunit of CCR4-NOT deadenylase, and its associated zinc finger protein 36-like 2 (ZFP36L2) protein were found to be essential for mRNA decay that accompanies oocyte meiotic maturation[6–8]. The B-cell translocation gene-4 (BTG4), which is an oocyte-specific adapter protein of CCR4-NOT, was identified as an MZT-licensing factor in mice that mediated mRNA clearance prior to ZGA[9–11]. These mechanisms comprise the currently known M-decay pathway in mice. In addition, terminal uridine transferase-4 and terminal uridine transferase-7 (TUT4/7)-mediated mRNA degradation not only maintained homeostasis of the maternal transcriptome during oogenesis, but also facilitated Z-decay in murine preimplantation embryos[12,13]. The maternal transcriptional coactivator YAP1 and its co-transcription factor TEAD4 were found to trigger the transcription of early zygotic genes, such as *Tut4/7*, and possibly genes encoding other unidentified mRNA destabilizers[14,15]. Further, these mechanisms comprise key components of the murine Z-decay pathway. Despite these findings in model animals of lower-level species, the dynamics of mRNA decay and mechanisms that govern stepwise maternal mRNA clearance during MZT in humans remain unclear.

Human preimplantation embryogenesis is a remarkably complicated, well-orchestrated process that relies on synchronization of oocyte maturation and zygotic genome activation (ZGA)[16,17]. Despite extensive research on murine as well as human oocytic and embryonic transcriptomes in recent years, many questions regarding key MZT events in humans remain unanswered[18]. For instance, in assisted human reproduction, the extent of cytoplasmic maturation of an oocyte is considered a determining factor for its developmental potential after fertilization[19,20]. However, whether the appropriate maternal mRNA degradation contributes to the cytoplasmic maturation of human oocytes and their developmental potential after in vitro fertilization (IVF) remains unclear. In addition, unlike mouse embryos, in which major ZGA is initiated at the 2-cell stage, human embryos undergo major ZGA at the 8-cell stage[21,22]. The proportion of human maternal transcripts with clearance that is ZGA-dependent remains undetermined. From a broader perspective, dysregulation of the maternal mRNA clearance process may be related to various disorders of the reproductive system, such as follicle growth retardation, oocyte maturation defects, early embryo arrest, oocyte aging, and ultimately, infertility[6,9,12,23,24]. Thus, investigating the stability regulation of maternal mRNAs during human MZT may facilitate the understanding of associated physiological, as well as pathological processes.

TUBB8 is a primate-specific β-tubulin isotype, the expression of which is confined to oocytes and the early embryo[25]. TUBB8 variants are genetic determinants of human oocyte maturation arrest that cause variable and mixed phenotypes in oocyte maturation and early embryo development[26,27]. However, whether the process of oocyte maturation-associated maternal mRNA decay was also disturbed in these mutated zygotes is unclear. In this study, we define and characterize the ZGA-dependent maternal mRNA clearance process during human MZT and demonstrate that subgroups of the human maternal transcripts are sequentially removed by M-decay and Z-decay pathways before and after ZGA. We also evaluate the association of maternal mRNA degradation defects with zygotic developmental arrest due to *TUBB8* mutations or unidentified reasons. These investigations aim to provide insight into the dynamics, functional importance, and pathological relevance of maternal mRNA decay during human MZT.

## Results

**Patterns of maternal mRNA degradation in human oocytes and embryos.** To identify patterns of maternal mRNA degradation during MZT in humans, we analyzed the degradation dynamics of human maternal mRNAs in GV oocytes, zygotes, and 8-cell embryos using published RNA-seq data (GSE101571)[28]. As illustrated in Fig. 1a, ZGA occurs at the 4–8-cell stage in the human embryo. Maternal mRNAs with reliable sequence annotations and with fragments per kilobase of transcript per million reads mapped (FPKM) of >2 at the GV stage (7271 genes) were selected. Those with significant decreases in mRNA levels between two stages at a magnitude of more than 2-fold were considered degraded maternal mRNAs and were classified into four clusters according to their degradation patterns: Cluster I (2372 genes), degraded from the GV stage to the zygote stage, and stable after fertilization; Cluster II (2259 genes), degraded from the zygote stage to the 8-cell stage; Cluster III (1109 genes), continuously degraded from the GV stage to the 8-cell stage; and Cluster IV (1531 genes), stable during MZT (Fig. 1a, b). To assess whether all Cluster IV transcripts remained stable beyond the 8-cell stage or if a subset of transcripts were degraded after this timepoint, we also analyzed transcript levels at the morula stage (Fig. 1c). This analysis indicated that only 176 of the 1531 Cluster IV transcripts were, in fact, degraded between the 8-cell and morula stage. The majority of Cluster IV transcripts remained stable between the 8-cell and morula stage. In human embryos, zygotic transcription activity is first detected at the 4-cell or 8-cell stage; thus, maternal mRNAs in Clusters II and III were considered candidates for ZGA-dependent decay, or Z-decay; further, Cluster I was considered a candidate for a maternally encoded mRNA decay pathway that acts before ZGA and is defined as M-decay.

We then asked whether maternal mRNA degradation in human embryos after fertilization depends on ZGA. Both in vitro controls and α-amanitin-treated human zygotes successfully developed to the 8-cell stage (Fig. 1a). These 8-cell embryos were collected for RNA-seq analysis. Significantly, transcripts of Clusters II and III were stabilized in α-amanitin-treated embryos (Fig. 1d). However, α-amanitin treatment only blocked the degradation of nearly half of the transcripts of Clusters II and III in mice. Thus, a ZGA-dependent mRNA decay pathway was found to operate during human MZT, in which it played a more important role than in mouse MZT.

In mice, long 3′-UTRs and high translational activity of Z-decay mRNAs conferred resistance to CCR4-NOT-mediated deadenylation during MZT, which showed that the length of the 3′-UTR is also a factor that determines mRNA stability[29,30]. We also observed in the human transcriptome that: (1) M-decay transcripts possessed shorter 3′-UTRs compared to Z-decay transcripts (Fig. 1e); (2) When multiple cytoplasmic polyadenylation element (CPE) and polyadenylation signal (PAS) were present in the 3′-UTR of mRNAs, they contributed to mRNA translation in an additive manner during oocyte maturation in

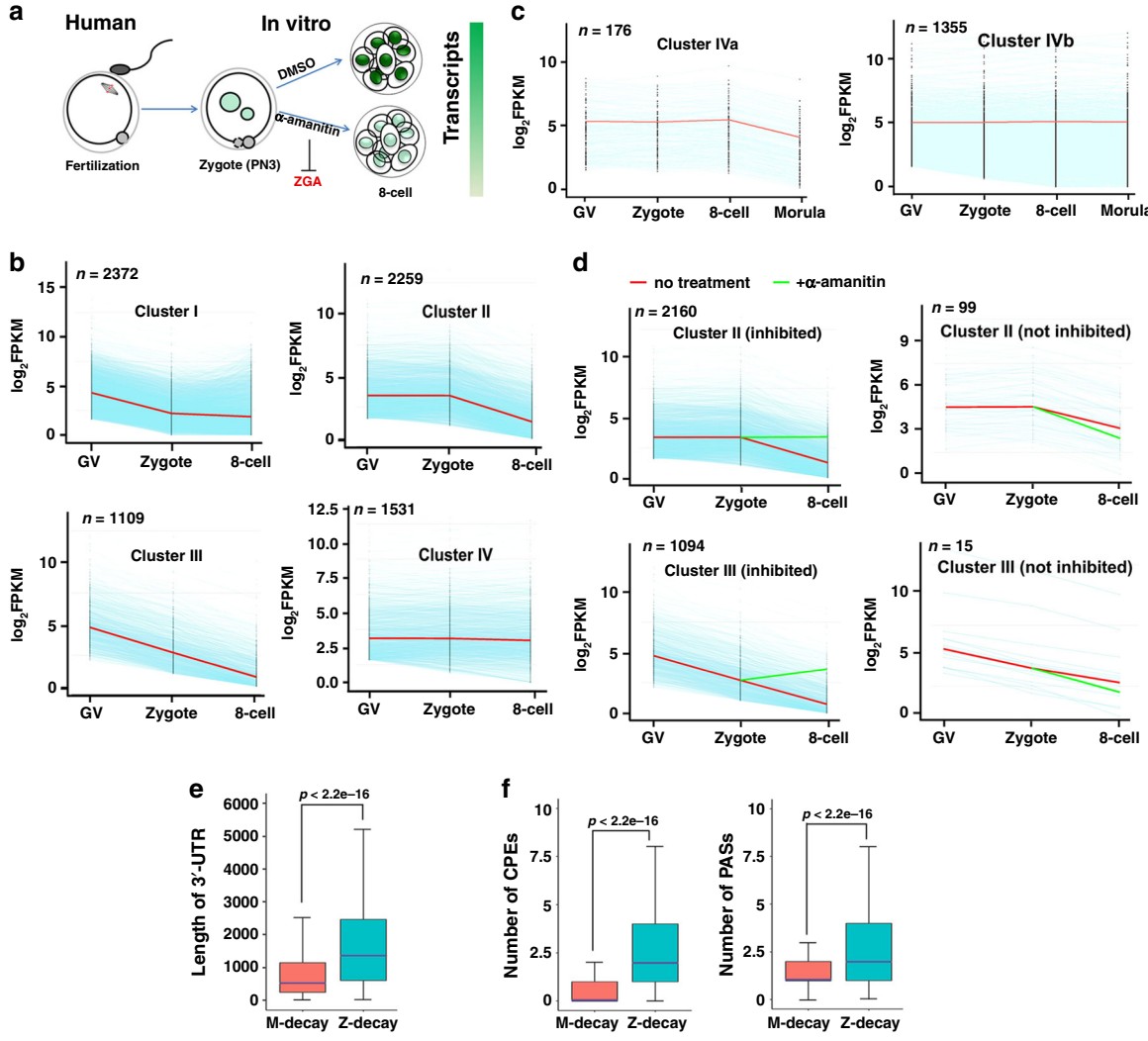

**Fig. 1 Dynamics of maternal mRNA clearance in human preimplantation embryos. a** This illustration shows the treatment of human oocytes and embryos for RNA-seq. Zygotes were treated with or without α-amanitin (25 ng/μl) and then cultured till they reached the 8-cell stage. **b** and **c** Degradation patterns of human maternal transcripts at the GV, zygote, 8-cell, and morula stages. Transcripts with FPKM > 2 at the GV stage were selected and further analyzed. Each light blue line represents the expression levels of one gene, and the middle red line represents the median expression levels of the cluster. **d** Degradation patterns of maternal transcripts in human embryos with or without α-amanitin treatment. Transcripts with FPKM (2-cell)/FPKM (zygote) < 0.5 were selected for analysis. Each light blue line represents the expression levels of one gene. The middle red line represents the median expression levels of the cluster. The green line represents the median expression levels of the cluster after α-amanitin treatment. **e** and **f** Average 3′-UTR length **e** and numbers of CPEs and PASs **f** in the 3′-UTR of the human M-decay and Z-decay transcripts. The box indicates the upper and lower quantiles, the purple thick line in the box indicates the median, and the whiskers represent the 2.5th and 97.5th percentiles. Data are presented as mean values ± SEM. *P* by a two-tailed Student's *t*-test. *n* = 2372 genes for M-decay; *n* = 2259 genes for Z-decay.

mice. Similarly, we found that, in humans, the numbers of CPEs and PASs were present in the 3′-UTRs of Z-decay mRNAs at 2 folds compared to those in the 3′-UTRs of M-decay mRNAs (Fig. 1f).

**Comparisons of human and mouse transcriptomes during the MZT.** To demonstrate the differences between human and mouse maternal transcriptomes, we directly compared the transcripts in human and mouse GV oocytes. This revealed that only half of the transcriptomes were overlapping (Fig. 2a), indicating that the homology between human and mouse maternal transcriptomes is low. Even fewer zygotically activated genes were shared by mouse 2-cell embryos and human 8-cell embryos (Fig. 2a). Maternal mRNAs were classed into four clusters according to the level change during MZT. We made comparisons of these four groups between mouse and human since they may utilize the same

mechanisms for maternal mRNA decay. Despite these four clusters being defined by similar criteria, the genes in each cluster were significantly different in human and mouse (Fig. 2b). Examples in Fig. 2c, d shows that some human M-decay transcripts were degraded by Z-decay pathways in mouse, and vice versa. Therefore, subsets of human mRNA might be regulated differently from mouse during the MZT.

**Classification of maternal mRNA degradation in human embryos.** To verify the RNA-seq data regarding mRNA dynamics during MZT in human embryos, we collected oocytes (GV and MII) and preimplantation embryos at the 8-cell and morula stages from volunteers (25–35 years old) for RNA extraction and quantitative RT-PCR (RT-qPCR). We detected levels of transcripts that were shown to be targets of BTG4 and CNOT6L in mice and were eliminated during oocyte meiotic maturation, i.e.,

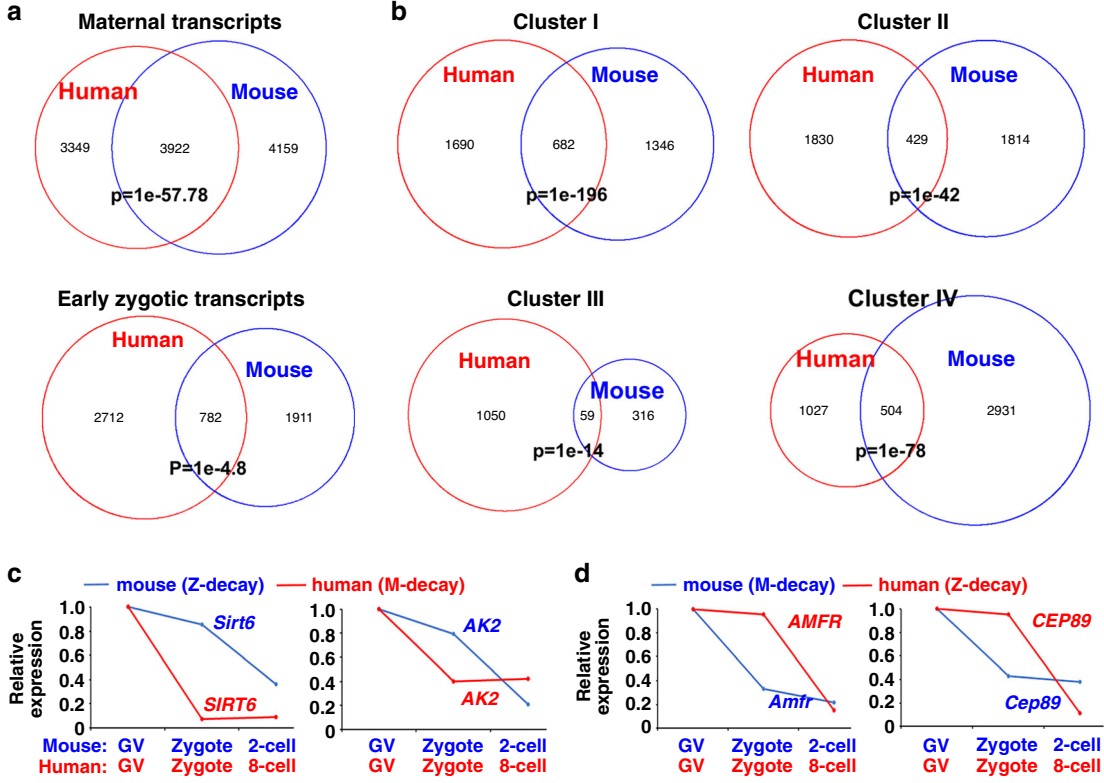

**Fig. 2 Comparisons of the maternal mRNA degradation pattern between mouse and human. a** Venn diagram showing the overlap of maternal and zygotic transcripts in GV oocytes and early embryos (mouse: 2-cell; human: 8-cell) from mouse and human. **b** Overlap of human and mouse maternal transcripts in Clusters I–IV that are classified in Fig. 1b. **c** and **d** Relative expression levels of representative transcripts in human and mouse oocyte/ embryos at the indicated stages.

M-decay. Single cell RT-qPCR results showed that these transcripts were also degraded during the GV-to-MII transition in human oocytes (Fig. 3a). Furthermore, the levels of these M-decay transcripts were comparable in MII oocytes and in zygotes, suggesting that degradation of these transcripts was largely completed by the MII stage and there is no significant degradation during the MII-to-zygote transition (Fig. 3a). Some transcripts that were eliminated by Z-decay were stable before fertilization and were degraded at the 8-cell stage (Fig. 3b). As in mice, the delayed removal of mRNAs encoding BTG4 and the catalytic subunits (CNOT7 and CNOT6L) of CCR4-NOT deadenylase, as well as PAN2 RNA deadenylase, was also observed in the human embryos: RT-qPCR and RNA-seq results showed that these transcripts were relatively stable until the 4–8-cell stage (Fig. 3c and d). BTG4 and CNOT7 proteins were undetectable in GV oocytes before meiotic maturation, but accumulated in maturing oocytes and in zygotes, as detected by immunofluorescence (Fig. 3e). Then these proteins decreased to basal levels at the 8-cell stage. This observation suggests that BTG4 and CCR4-NOT may play roles in human maternal mRNA decay, whereas they themselves were degraded until other maternal mRNAs were eliminated.

We also examined the potential ZGA factors that were involved in the Z-decay pathway in human embryos. The transcription factor TEAD4 was zygotically expressed in mice and was required for Z-decay in preimplantation embryos[15,30]. Similarly, *TEAD4* transcription in human embryos was activated at as early as the 4-cell stage, and its mRNA levels increased 4-fold from the 8-cell stage to the morula stage, as determined by RNA-seq results (Fig. 3f)[28]. In contrast, the mRNA levels of *YAP1* were relatively stable during MZT (Fig. 2f). Immunofluorescence results showed that YAP protein evenly distributed in the human GV oocytes

and zygotes, but accumulated in the nuclei of 8-cell embryos (Fig. 3g). The 3′-terminal uridylyl transferase 4 and 7 (TUT4/7)-dependent mRNA 3′-oligouridylation in mice participated in mRNA decay and sculpted the maternal transcriptome[12]. *TUT7*, which is the downstream factor of *TEAD4*[30], was also expressed in human oocytes at higher levels compared to *TUT4*, but maternal *TUT7* transcripts were removed during oocyte maturation and fertilization. Nevertheless, transient expression of zygotic *TUT4/7* was detected from the 8-cell stage to the morula stage, with *TUT7* levels being higher than those of *TUT4* (Fig. 3f). The *TUT7* expression levels peaked at the 8-cell stage and then rapidly decreased at the morula stage. The expression window of the human zygotic *TUT4/7* gene overlapped with the time frame of Z-decay.

**The *TUBB8* mutation did not affect M-decay in humans**. In a clinical context, normal embryos should develop at the 8-cell stage 3 days after IVF[31]; however, there were embryos that were fertilized, as evidenced by the formation of pronuclei, that remained arrested at the 1-cell stage[32]. Zygotic arrest of some embryos was due to gene mutations, such as *TUBB8* mutations that affect cell division[33].

To investigate whether mRNA degradation was blocked by *TUBB8* mutation-induced cell cycle arrest, we microinjected mRNA encoding mutated TUBB8 (TUBB8[V255M], which is a dominant negative mutant[34]) into mouse oocytes (Fig. 4a). Consistent with the results of previous studies, oocytes that overexpress TUBB8[V255M] had a GVBD rate similar to that of control oocytes that overexpress wild-type TUBB8; however, the PB1 emission rate of oocytes that overexpress TUBB8[V255M] was significantly lower than that of the control group (Supplementary

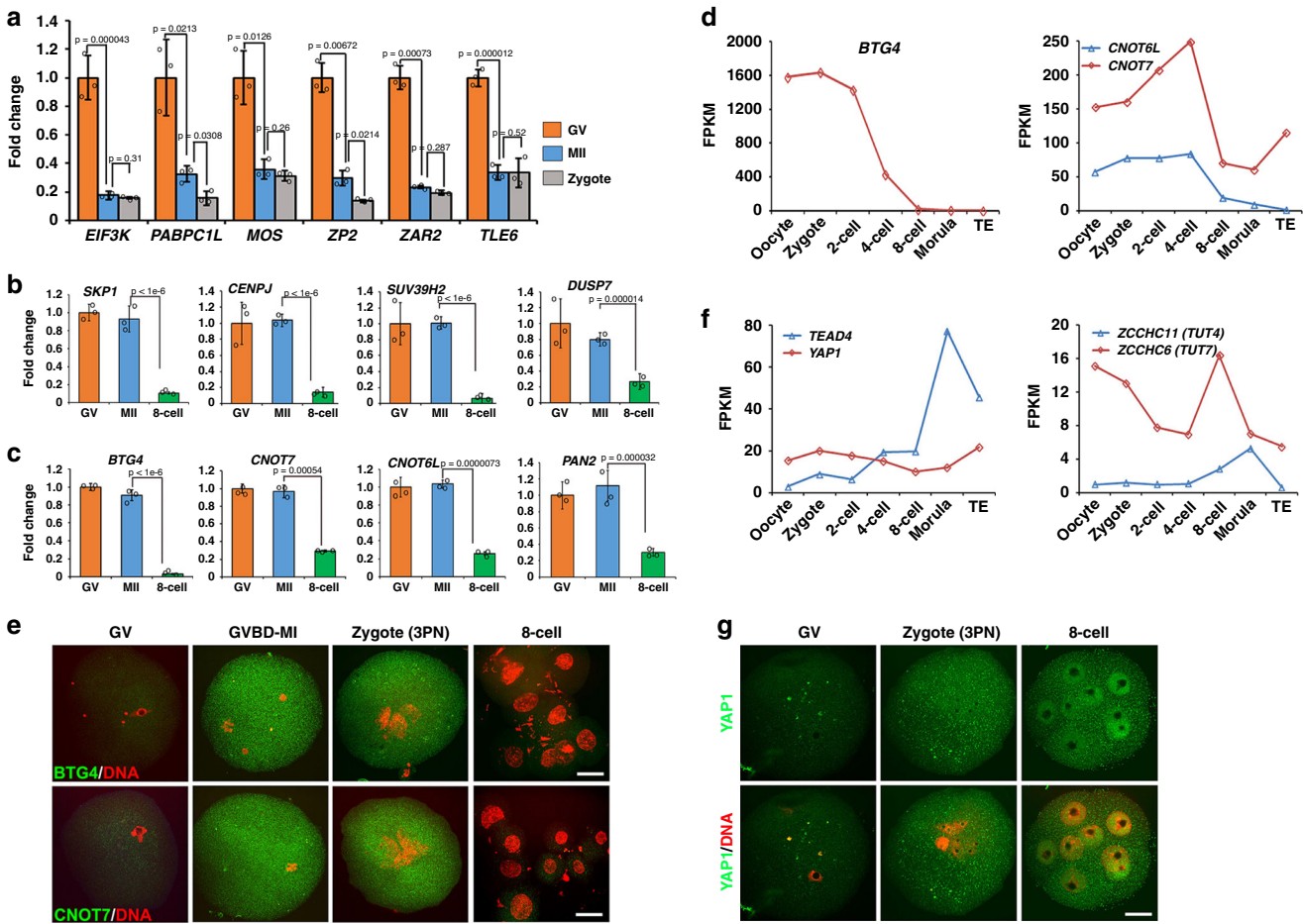

**Fig. 3 mRNA dynamics of indicated genes during MZT in human. a** RT-qPCR results showing the mRNA levels of selected M-decay transcripts at the GV, MII, and zygote (3PN) stages. **b** and **c** RT-qPCR results showing the mRNA levels of selected Z-decay transcripts at the indicated stages. Data are presented as mean values ± SEM. *P* by a two-tailed Student's *t*-test. *n* = 3 biological replicates. **d** and **e** The RNA-seq results showing the levels of transcripts encoding key factors of the M-decay **d** and Z-decay **e** pathways in human oocytes and early embryos. FPKM values were extracted from previously published data (GSE36552[28]). **f** and **g** Immunofluorescence results showing the protein levels of BTG4, CNOT7, and YAP in human oocytes, zygotes containing three pronuclei (3PN), and 8-cell embryos. Scale bars = 40 μm. Immunostaining of each antibody was independently repeated for three times with similar results.

Fig. 1a). Oocytes that overexpress TUBB8$^{V255M}$ failed to assemble bipolar spindles and were arrested at the pre-MI stage. Further, chromosomes were not aligned at the equatorial plates (Fig. 4b). Although 40% of TUBB8$^{V255M}$-overexpressing oocytes were found to release PB1 and reached the MII stage, they did not assemble bipolar MII spindles (Fig. 4b). We then detected M-decay of maternal mRNAs in these oocytes. RT-qPCR results showed that degradation of indicated mRNAs was not affected, regardless of whether the PB1 was released (Fig. 4c).

We also detected the M-decay of human mRNAs in zygotes that were derived from *TUBB8*-mutated oocytes. Samples (1 and 2 from *TUBB8*$^{V255M}$ patients; 3 and 4 from *TUBB8*$^{G308S}$ patients) were collected, as shown in Fig. 5a and b. RT-qPCR results showed that the indicated mRNAs in the arrested zygotes carrying maternal *TUBB8* mutations were at comparable levels with those in 3PN zygotes, suggesting that they are normally degraded in maternal *TUBB8* mutated zygotes (Fig. 5c and Supplementary Fig. 1b). These observations are consistent with those of previous reports suggesting that M-decay in mouse oocytes is not affected by nocodazole-induced meiotic spindle disruption[35].

Collectively, these results indicated that the arrest of meiotic cell division did not affect M-decay of maternal transcripts. If mRNA degradation defects were found in some arrested human

embryos, these defects were not considered secondary consequences of spindle assembly abnormalities of the oocytes.

**M-decay was frequently impaired in development-arrested human zygotes.** To verify whether the defects of maternal mRNA degradation were causes of human preimplantation embryo arrest, we profiled the transcriptome of arrested 1-cell embryos that were derived from seven mutation-unidentified (unid) patients at day 3 after IVF. Since it is ethically difficult to collect normal human zygotes for this specific experimental purpose and since the *TUBB8* mutation did not affect M-decay of known maternal transcripts, arrested 1-cell embryos from two *TUBB8*-mutated patients (V255M and G308S) were used as controls for single-cell RNA-seq.

The gene expression levels were assessed by FPKM. A principal component analysis revealed that two samples from *TUBB8*-mutated patients had high correlations, and five out of seven samples from unid-patients also had high correlations (Fig. 6a). A heatmap also showed high correlations among five samples from unid-patients (average *r* = 0.900; Supplementary Fig. 1c, f and Supplementary Table 1), whereas these five samples were significantly different from the two samples of *TUBB8*-mutated patients (Fig. 6a, Supplementary Fig. 1c and Supplementary

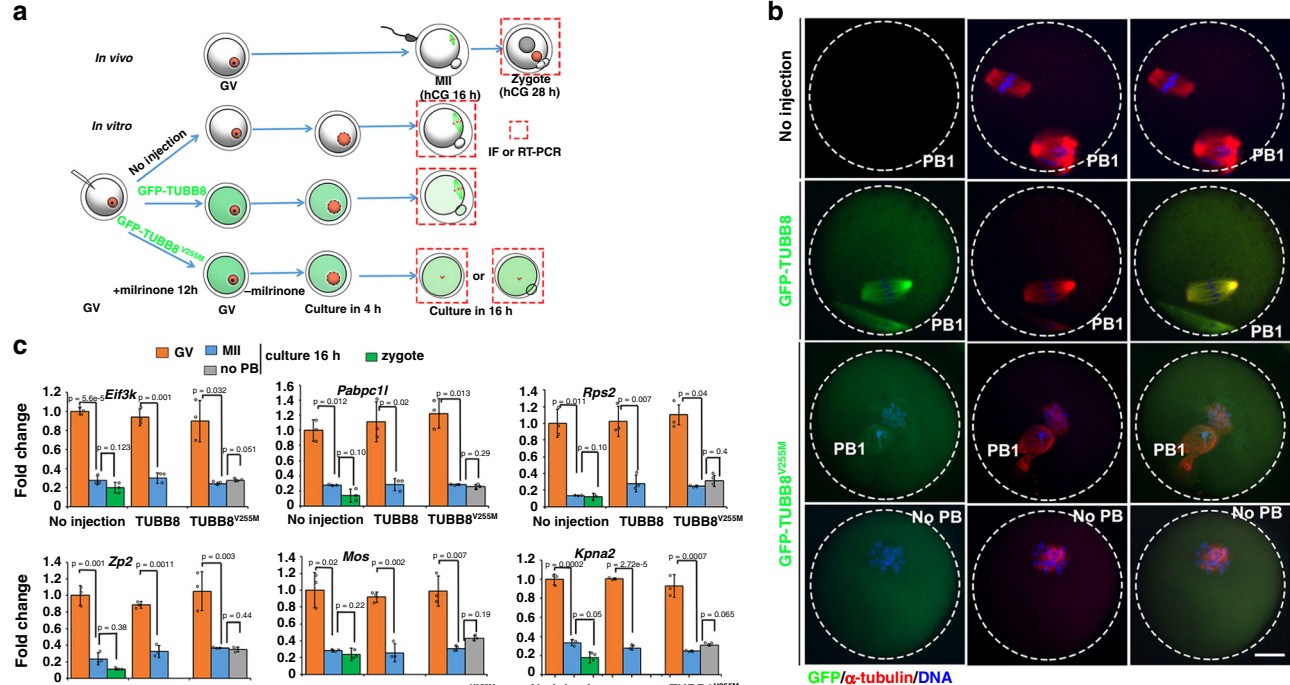

**Fig. 4 TUBB8 mutation did not affect M-decay in mouse oocytes. a** A diagram of spindle assembly disruption after microinjection of mRNAs encoding GFP-TUBB8$^{V255M}$. **b** Confocal microscopy results showing spindle assembly of mouse oocytes at 16 h after meiotic resumption. GV oocytes were microinjected with mRNAs encoding either wild type or mutant (V255M) TUBB8 and then cultured, as shown in **a**. Endogenous α-tubulin was detected by immunofluorescence (red). DNA was labeled by 4',6-diamidino-2-phenylindole (DAPI). Scale bars = 20 μm. The immunostainings were independently repeated for three times with similar results. **c** RT-qPCR results showing the relative levels of M-decay transcripts in mouse oocyte microinjected with mRNAs encoding wild type or mutant (V255M) TUBB8, as shown in **a**. Levels of these transcripts were also detected in normal zygotes as controls. n = 3 biological replicates. Data are presented as mean values ± SEM. P by two-tailed Student's t-tests. n.s. non-significant, No PB no polar body.

Table 1). These observations indicate that the five embryos from unid-patients may have been arrested for the same reason, but not due to meiotic division defects like the *TUBB8*-mutated oocytes.

It was found that 3712 and 2493 transcripts were upregulated and downregulated more than 2-fold in the five unid-patient embryos with high transcriptome correlations (Fig. 6b, c). More transcripts were upregulated than they were downregulated in these embryos when we increased the thresholds of the analyses (Fig. 6d, e). The #6 unid-patient also displayed remarkable maternal mRNA accumulation, as more genes were upregulated than they were downregulated in the embryos of this patient compared to control embryos (Supplementary Fig. 1d). However, relatively small numbers of transcripts in the #7 unid-patient were upregulated or downregulated (Supplementary Fig. 1e), and the numbers of upregulated and downregulated genes were not very different (646 versus 494). Overall, mRNA clearance was impaired in 6 of 7 unid-patients.

Since the transcriptomes of the #1–5 embryos had high correlations, they were used for further analyses. Nearly 50% (1490 in 3712) of the transcripts that were upregulated in the unid-patient embryos should have been degraded in normal embryos during the GV-to-zygote transition (i.e., Cluster I and III of Fig. 1b) (Fig. 6f). Among mRNAs degraded in normal embryos during the GV-to-zygote transition, only 1431 of 3179 were degraded in unid-patient embryos (Fig. 6g); nearly 50% (1490/3179) were stabilized in unid-patient embryos. In contrast, only <10% (285/3179) were downregulated in unid-patient embryos. We further performed gene ontology (GO) analyses on the transcripts that are upregulated in the arrested embryos with fold changes of >2. Transcripts that are related to translation-related functions (red bars) and mRNA stability (green bars) were

enriched (Fig. 6h), and they may have caused over-translation of the accumulated maternal mRNAs and led to cell division defects.

**M-decay defects potentially cause embryo arrest in humans**. To further determine whether M-decay defects associated with early embryo arrest in humans, we collected arrested zygotes from 4 *TUBB8*-mutated patients and 15 unid-patients and verified the levels of known M-decay transcripts by RT-qPCR (Fig. 7a, b). The mRNA levels of indicated transcripts were consistently low in the samples from *TUBB8*-mutated patients (Fig. 7a, b). In zygotes from 8 unid-patients (#1–8), at least 4 out of 6 detected transcripts showed significant accumulation compared to that in the maternal *TUBB8*-mutated zygotes, suggesting that M-decay was defective (Fig. 7a and Supplementary Table 2). In contrast, in the other 7 unid-patients (#9–15), these M-decay transcripts were not synergistically upregulated, suggesting that these embryos were arrested due to reasons other than M-decay defects (Fig. 7b and Supplementary Table 2).

Recent studies have indicated that the oocyte-expressed MZT licensing factor BTG4 mediates maternal mRNA degradation in mouse oocytes and zygotes by recruiting the CCR4-NOT complex to transcripts that undergo active translation[9–11]. Murine CNOT6L, which is a CCR4-NOT catalytic subunit, is required for meiosis-coupled maternal mRNA decay[6]. RT-qPCR results indicate that the *CNOT6L*, *CNOT7*, and *BTG4* expression was significantly lower in human embryos that have M-decay defects compared to that in *TUBB8*-mutated embryos, whereas the decrease in *CNOT6L*, *CNOT7*, and *BTG4* levels was less remarkable in the arrested embryos of unid-patients without M-decay defects (Fig. 7c, d and Supplementary Table 2). These results suggest that similar to the mechanisms of the mouse MZT,

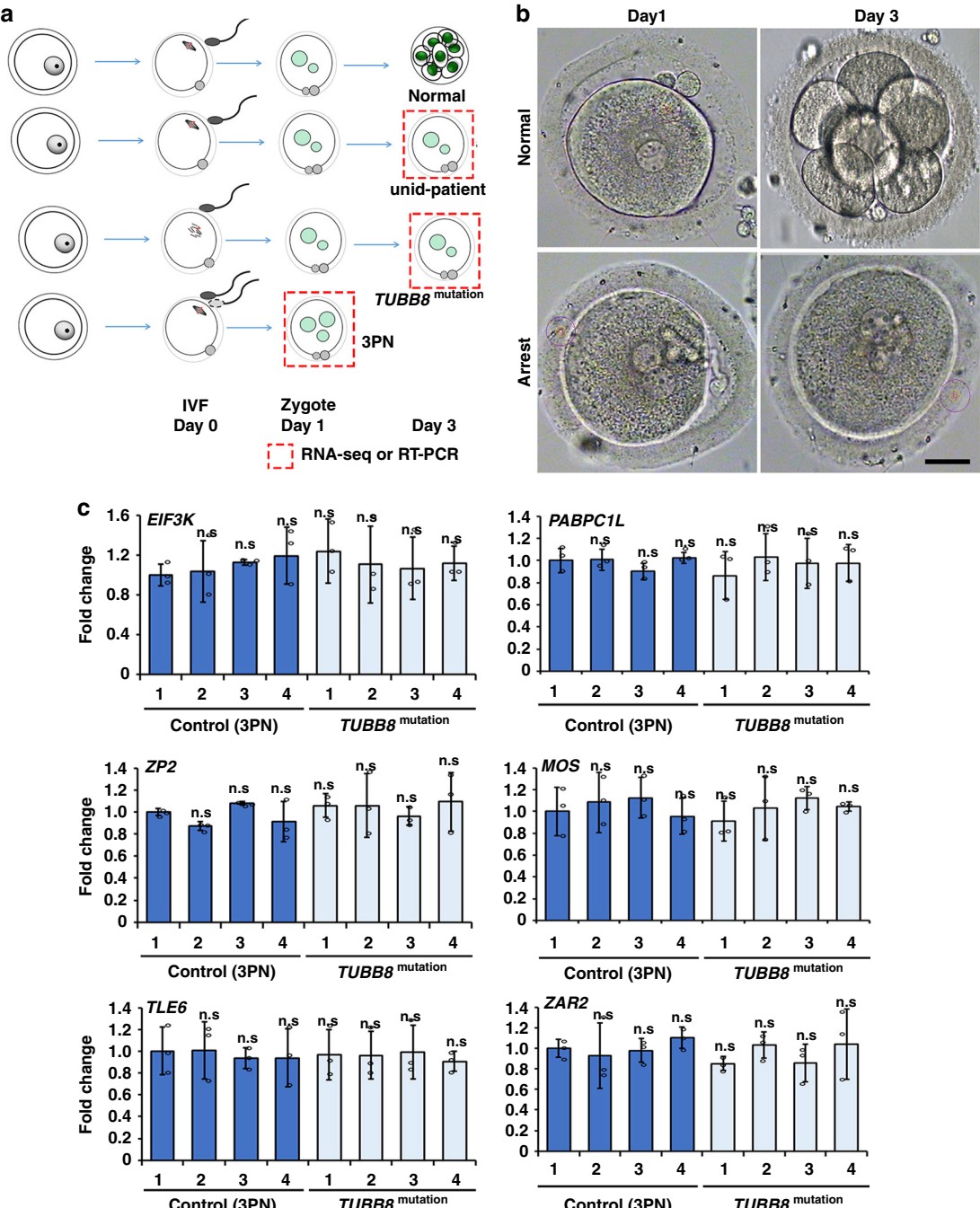

**Fig. 5 *TUBB8* mutation did not affect M-decay in human zygotes. a** A illustration of human embryo collection for RT-qPCR or RNA-seq. Oocytes that were derived from unidentified or *TUBB8*-mutated patients were in vitro fertilized and cultured for 3 days before RNA isolation. **b** Representative images showing morphologically normal and 1-cell stage-arrested embryos at 3 days after IVF. Scale bars = 40 μm. All observed normal (*n* = 4) and arrested embryos (*n* = 15) looked like this in three independently repeated observations. **c** RT-qPCR results showing the relative levels of M-decay transcripts in the 3PN zygotes 1 day after IVF and arrested zygotes of *TUBB8*-mutated patients 3 days after IVF. 1–4 represent embryos from different women. Data are presented as mean values ± SEM. n.s. non-significant by one-way ANOVA. *n* = 3 independent experiments.

BTG4 and CCR4-NOT may also participate in the M-decay pathway of human embryos.

**Human embryos with ZGA defects also have defects to remove maternal transcripts through Z-decay.** ZGA have been shown to occur at the 8-cell stage in human embryos, but the association between Z-decay and the developmental potential of early embryos has never been assessed in human. In a clinical context, normal embryos should develop into blastocysts 5 days after IVF.

However, some embryos reached the 8-cell stage but fail to form blastocysts. Thus, we compared the transcriptomes of normal and 8-cell stage-arrested human embryos. Morphologically normal embryos at the 1-cell and 8-cell stages were collected. The 8-cell arrested embryos at day 5 after IVF were separately collected from six patients who experienced repeated developmental failure of preimplantation embryos after IVF (Fig. 8a, b). Individual normal and arrested embryos were subjected to single-embryo RNA-seq analyses, and the gene expression levels were assessed by FPKM.

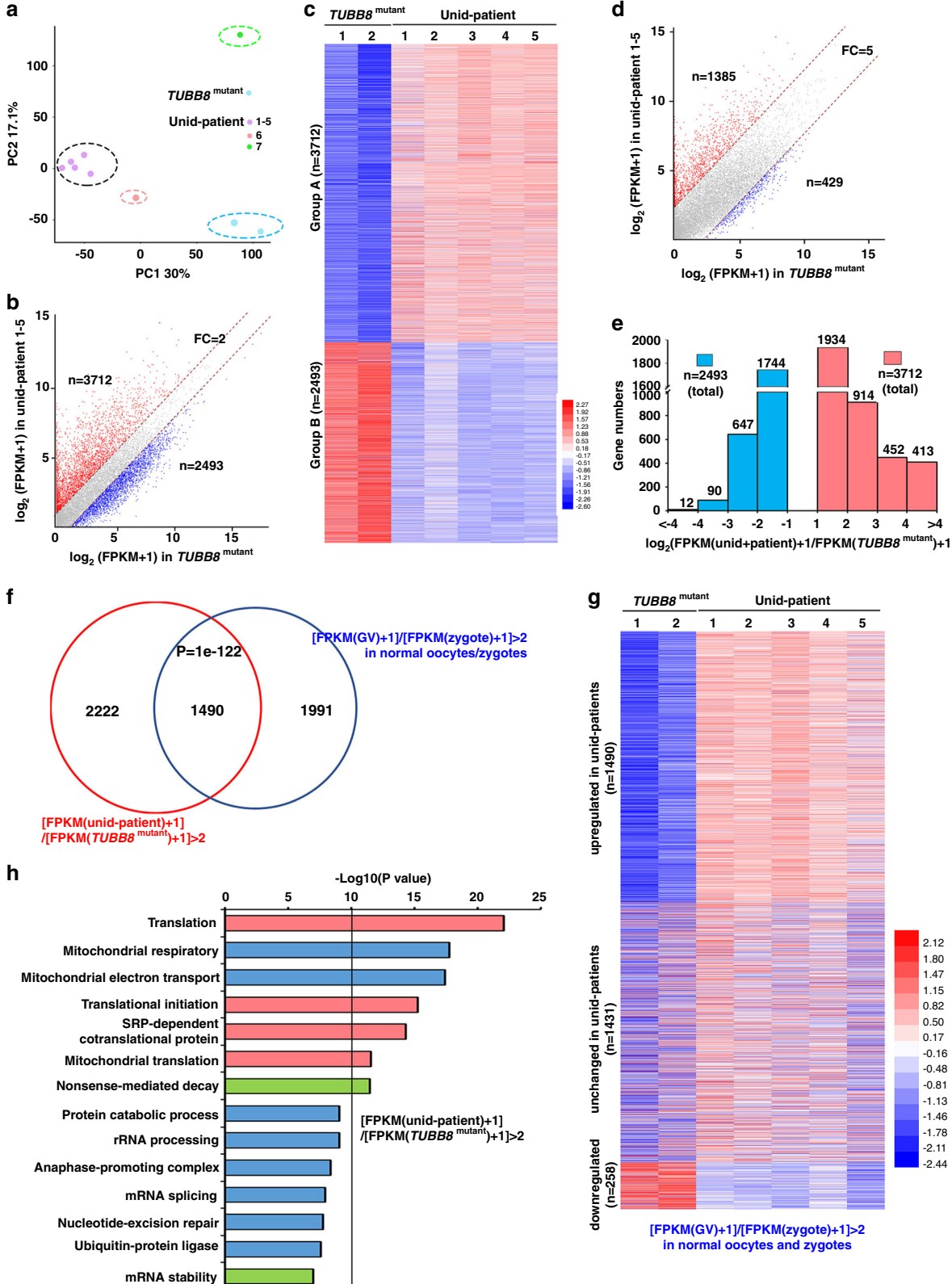

All three normal embryo samples at the 1-cell or 8-cell stages had high correlations following principal component analysis, whereas 4 (#1, 4–6) of 6 arrested 8-cell embryos had high correlations (Fig. 8c). A heatmap also showed that these four samples had high correlations (average $r = 0.748$) and significantly differed from three normal 8-cell embryos (Supplementary Fig. 2a and Supplementary Table 3). In the following experiment,

RNA-seq results of these four samples were further compared to those of normal embryos. There were 2968 and 5297 transcripts that were downregulated and upregulated more than 2 folds in arrested embryos when compared to normal 8-cell embryos (Fig. 8d). When the threshold of the fold changes was increased to 5, there were more genes that were upregulated than those that were downregulated (Fig. 8e and Supplementary Fig. S2b). In the

**Fig. 6 Transcriptome analyses for developmental arrested human zygotes after IVF. a** The principal component analysis (PCA) results of developmental arrested zygotes 3 days after IVF. *TUBB8*mutant indicates zygotes of *TUBB8*-mutated patients; unid-patient indicates mutation-unidentified patients. **b** A scatter plot is shown, which compares transcripts in arrested zygotes that were derived from *TUBB8*-mutated and unid-patients. Transcripts decreased or increased more than 2 folds in unid-patient samples compared to *TUBB8*mutant samples, which are highlighted in blue and red, respectively. *n* gene number, FC fold change; 1–5, embryos from different unid-patients. **c** Heatmap of genes upregulated or downregulated more than 2 folds in zygotes from unid-patients compared to *TUBB8*mutant samples. The definition of Groups A and B is described in the text. **d** A scatter plot is shown, which compares transcripts in arrested zygotes that were derived from *TUBB8*-mutated and unid-patients. Transcripts decreased or increased more than 5 folds in zygotes of unid-patients compared to *TUBB8*mutant samples, which were highlighted in blue and red, respectively. *n* gene number; FC fold change; 1–5, represent embryos from different unid-patients. **e** Numbers of transcripts that were upregulated (red) or downregulated (blue) in arrested zygotes derived from unid-patients compared to those derived from *TUBB8*-mutated patients. **f** Venn diagram showing the overlap of upregulated transcripts in arrested zygotes of unid-patients and the degraded transcripts from the GV-to-zygote transition in normal oocytes. $P = 1e-122$ by a two-tailed Student's *t*-test. **g** Heatmap showing the levels of M-decay transcripts (downregulated more than 2 folds from the GV stage to the zygote stage in normal samples (FPKM (GV) + 1/FPKM (zygote)+1) >2)) in arrested zygotes from *TUBB8*-mutated patients and unid-patients. **h** Gene ontology analysis of transcripts upregulated or downregulated more than 2 folds in arrested zygotes derived from unid-patients compared to those from *TUBB8*-mutated patients.

arrested embryos #2 and #3, however, the numbers of upregulated and downregulated genes were comparable (Supplementary Fig. 2c, d). Thus, mRNAs were accumulated in the arrested embryos #1, #4, #5, and #6.

A gene set enrichment analysis of the 2926 downregulated transcripts in the arrested embryos revealed that 1373 of these (~50%) belonged to early zygotically expressed genes of normal embryos (Fig. 8f). Thus, ZGA was at least partially impaired in these embryos. In addition, among the 4074 Z-decay transcripts that were detected in normal embryos, 775 transcripts were stabilized in the arrested embryos (Fig. 8g). Among these transcripts, 223 belonged to the previously identified ZGA-dependent Z-decay transcripts (Fig. 8g). A heatmap showed that the changed transcriptomes of four arrested embryos were consistently and significantly different from those of normal embryos (Fig. 8h). Further, a GO analysis revealed that the genes that failed to be expressed in arrested embryos were primarily associated with genome transcription and mRNA splicing (Fig. 8i). In contrast, the maternal transcripts that were accumulated in the arrested embryos were associated with the cell cycle, maternal behavior, and protein ubiquitination (Fig. 8j). These results were consistent with the phenotype of the prolonged 8-cell stage in these embryos.

**Z-decay defects were detected in the early development arrested human embryos**. We next investigated whether the Z-decay defects are frequently associated with the 8-cell arrest of human embryos. Embryos that were arrested as 8-cell embryos were separately collected at day 5 after IVF from 14 patients who experienced repeated preimplantation developmental failure. In a high proportion of these embryos, mRNA expression levels of the key ZGA factor MYC (12/14) and Z-decay factors (10/14 for *TEAD4* and 9/14 for *TUT7*), except for *TUT4*, were significantly lower than normal (Fig. 9a, b and Supplementary Table 4). Although these 8-cell arrested embryos had developed 2 days longer than the control embryos before RNA extraction, the maternal transcripts that were known to be degraded through the Z-decay pathway remained at higher levels in these embryos than in the normal 8-cell embryos. These include factors that are associated with the cell cycle (*CENPJ*), protein degradation (*SKP1*), maternal mRNA degradation (*CNOT7*), and histone H3 methylation (*SUV39H2*) (Fig. 9c, d and Supplementary Table 4)[36–39]. These results indicate an association between the Z-decay of maternal mRNAs and preimplantation developmental competence of human embryos.

**Inhibition of YAP–TEAD4 activity impaired *TUT4/7* expression and Z-decay in mouse and human embryos**. In the following experiments, we aimed to repress the YAP–TEAD4 activity in early human embryos using verteporfin, a small molecule that prevents the YAP–TEAD4 interaction, and then determine if the zygotic expression of *TUT4/7* and removal of Z-decay transcripts were impaired. We first confirmed the effects of verteporfin treatment in mouse. When GV oocytes were cultured in medium containing 1 µM verteporfin, meiotic maturation were as normal as the control group (Supplementary Fig. 3a, b), suggesting that the molecule is not toxic at this concentration. However, zygotes cultured at the presence of verteporfin had lower developmental rates than the control zygotes, with significant arrest at the 2–4-cell stages (Supplementary Fig. 3c, d). In verteporfin-treated 2-cell embryos, expression of known YAP–TEAD4 target genes, including *Tut4/7*, was repressed (Supplementary Fig. 3e). Meanwhile, the known mouse Z-decay transcripts accumulated in these embryos (Supplementary Fig. 3f). These phenotypes were similar to those observed in maternal *Yap1* knockout or TEAD4-inhibited embryos, suggesting that verteporfin effectively inhibited YAP–TEAD4 activity in cultured embryos.

Next, we cultured human zygotes with 3PN in medium containing verteporfin. The 3PN zygotes are usually caused by polyspermic IVF and are ethically approved to be used for research purpose. Approximately 40% 3PN zygotes developed to the 8-cell stage, with or without verteporfin treatment (Fig. 10a). These 8-cell embryos were collected for RT-qPCR analyses. The results showed that *TUT4/7* expression was repressed (Fig. 10b, c), whereas representative Z-decay transcripts accumulated (Fig. 10d, e) in the verteporfin-treated embryos. Therefore, YAP–TEAD4 is likely to have a conserved function to trigger zygotic *TUT4/7* expression as well as Z-decay transcript removal in both mouse and human early embryos.

**Discussion**

Studies in model systems has shown that both maternal and zygotic transcript degradation pathways are functional in the early mouse embryo during MZT[4,5,30]. When M-decay was impaired in mice, the embryos were arrested at the 1–2-cell stages, whereas Z-decay is required for mouse embryo development beyond the 4-cell stage[30,40]. However, whether mRNA decay (including M-decay and Z-decay) also plays a key functional role in human embryo development has not been investigated until this study. Thus, maternal mRNA decay defects have never been associated with early developmental arrest of human embryos after IVF. The current data mainly provided correlative rather than causal evidence that the factors facilitating mouse maternal mRNA decay may also be involved in the regulation of maternal mRNA stability during human MZT. Meanwhile, in another study we have identified infertile women carrying *BTG4*

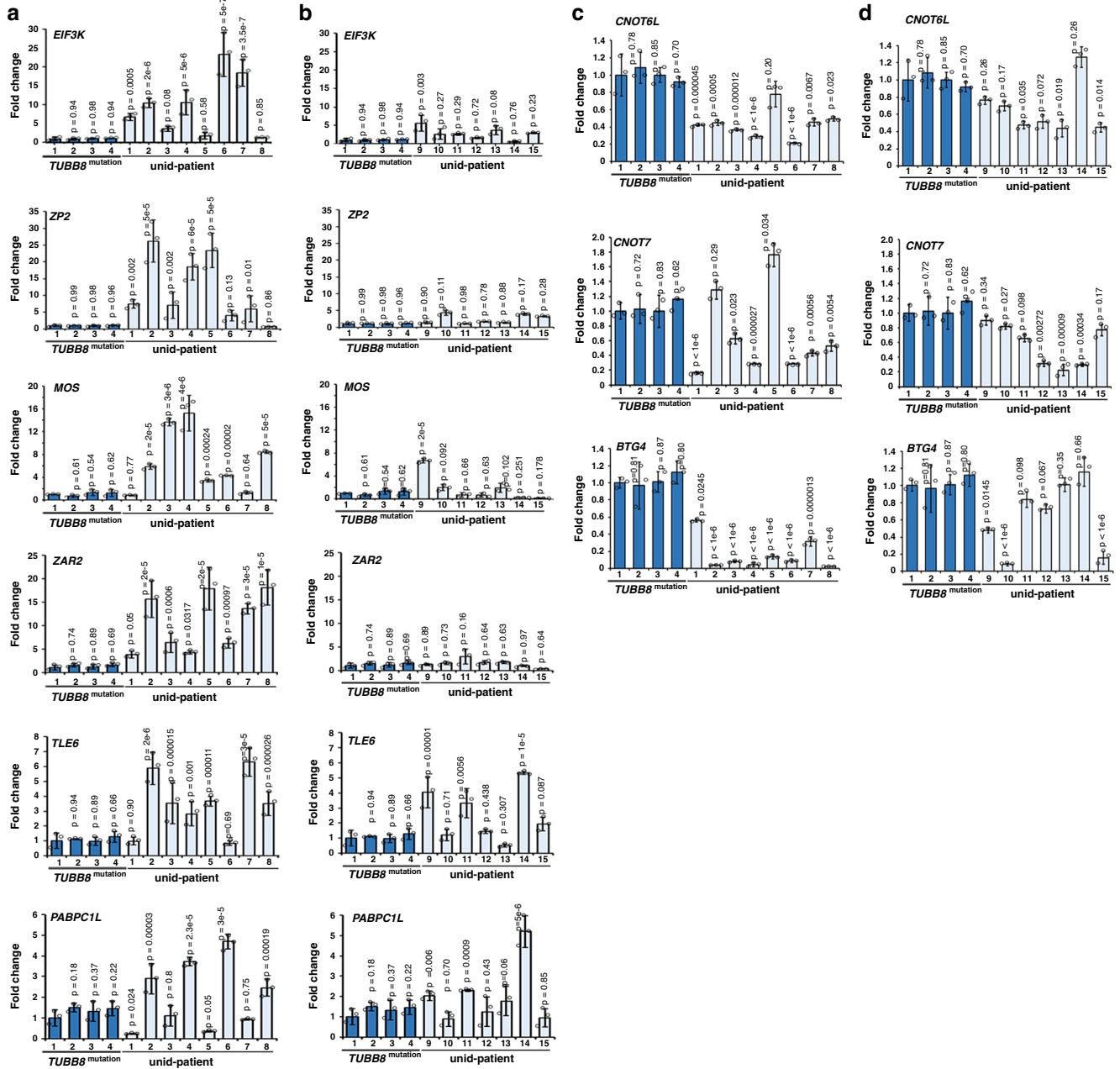

**Fig. 7 Levels of maternal transcripts in development-arrested human zygotes. a** and **b** RT-qPCR results showing the mRNA levels of selected M-decay transcripts in arrested zygotes derived from *TUBB8*-mutated and unid-patients 3 days after IVF. **c** and **d** RT-qPCR results showing the mRNA levels of *BTG4, CNOT7,* and *CNOT6L* in arrested zygotes that were derived from *TUBB8*-mutated and unid-patients. Data are presented as mean values ± SEM. *P* value by one-way ANOVA. $n = 3$ independent experiments.

mutations. The zygotes from these women were arrested at the 1-cell stage and exhibited defects in maternal mRNA degradation. The phenotypes were similar to those we have observed in *Btg4* knockout mice[41]. The identification of *BTG4* mutations in infertile women supports our hypothesis that BTG4/CCR4-NOT-induced mRNA deadenylation is involved in the regulation of maternal mRNA stability during human MZT. Also consistent with our working model, *TUT4/7* expression and Z-decay of maternal transcripts was impaired in human 8-cell embryos derived from 3PN zygotes, when YAP–TEAD4 activity was inhibited. These results provide evidence that YAP and TUT4/7 are likely regulating Z-decay of maternal mRNA during human MZT.

In this study, oocyte and embryo transcriptomes of human and mouse origin, as well as human embryo transcriptomes generated by different groups were compared. The absolute FPKMs can vary among different datasets due to differences in input RNA quantity, the efficiency of reverse transcription, and detection sensitivity. Furthermore, human samples obtained from the clinic often vary significantly across many factors, including patient age, genetic background, living environment, diet, and other factors. These factors cannot be strictly controlled as they are in experiments using a mouse model. Therefore, it is common to observe fewer overlaps of transcriptomic datasets published by different groups. The actual overlapped genes in most analyses of this study should be more than it appeared.

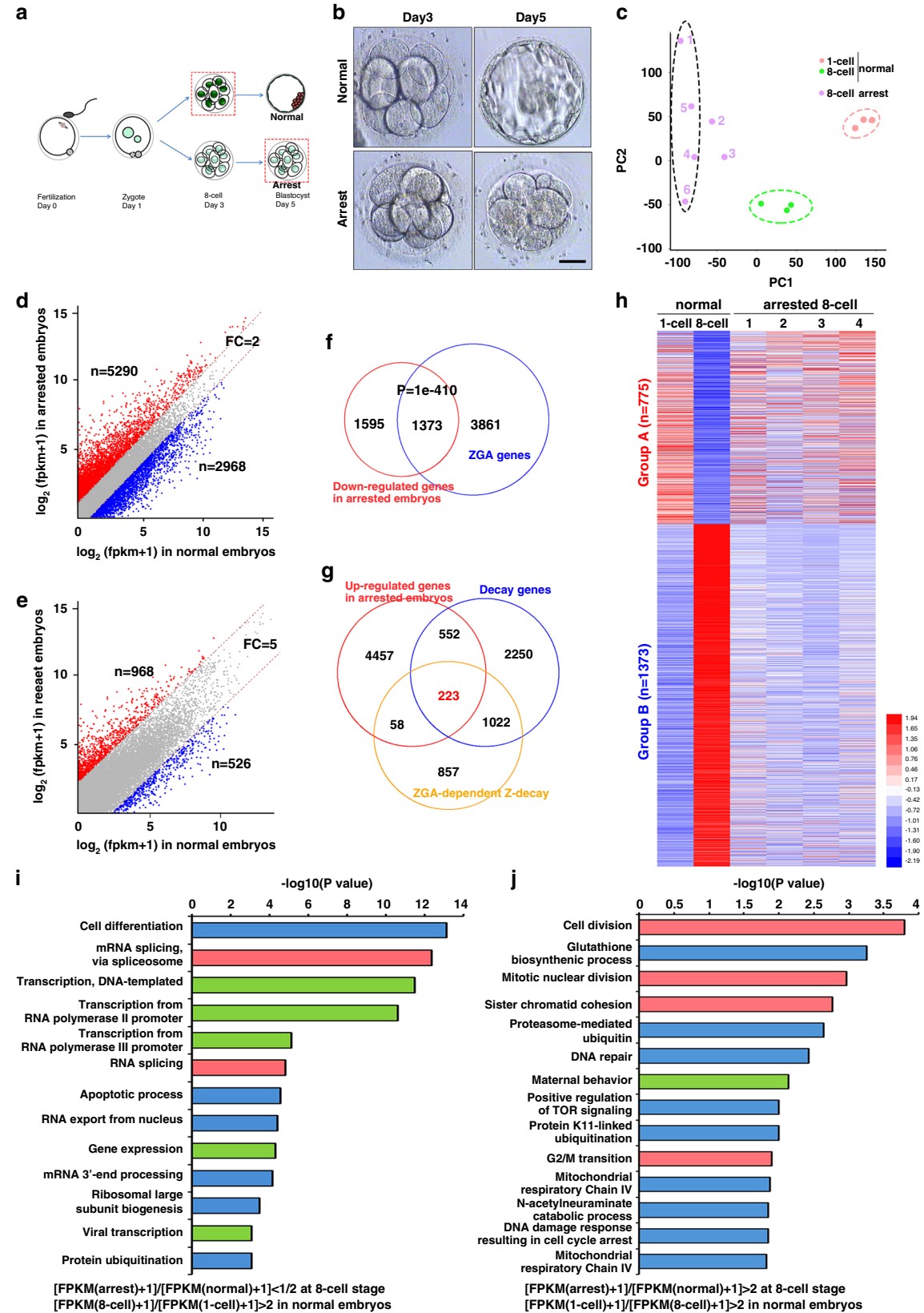

It is likely that, in addition to mRNA degradation, the maternal proteins are also removed for embryo development[42–44]. The results of our GO analyses suggest that the zygotes failed to undergo normal M-decay of maternal mRNAs, which indicates that the transcripts related to protein translation and mRNA stability were enriched and may be the cause of over-translation of the accumulated maternal mRNAs eventually leading to cell division defects[6,45]. Similarly, due to ZGA defects, many of the maternal mRNAs that were associated with the meiotic cell cycle and that should have been degraded by the Z-decay pathway were

**Fig. 8 Transcriptome changes in 8-cell stage-arrested human embryos. a** The illustration shows the collection of human embryos to perform RNA-seq. Zygotes obtained by IVF were cultured to the 8-cell or blastocyst stages. **b** Representative images of morphologically normal and 8-cell stage-arrested embryos. Scale bars = 50 μm. **c** The PCA results of embryos at the indicated stages. Arrested 8-cell embryos that were derived from unidentified patients were in vitro fertilized and cultured for 5 days before RNA isolation. 1–6, represent embryos from different unid-patients. All observed normal (n = 3) and arrested embryos (n = 14) looked like this in three independently repeated observations. **d** and **e** are derived from unid-patients (#1,4–6) to that of normal 8-cell embryos. Transcripts decreased or increased by more than 2 folds **d** or 5 folds **e** in arrested embryos compared to normal embryos, which were highlighted in blue or red, respectively. n, gene number; FC fold change. **f** Venn diagrams show the overlap of down-regulated transcripts in arrested embryos and ZGA transcripts in normal embryos. P = 1e−410 by a two-tailed Student's t-test. **g** Venn diagrams show the overlap of up-regulated transcripts in arrested embryos, the degraded transcripts from the zygote stage to the 8-cell stage in normal embryos, and ZGA-dependent Z-decay transcripts during normal MZT. P = 1e−14 by a two-tailed Student's t-test. **h** A heatmap illustration shows differentially expressed transcripts in normal and arrested embryos. Group A, transcripts that are degraded (fold change >2) during the zygote-to-8-cell transition in normal embryos, but that remained stable in the arrested 8-cell embryos. Group B, transcripts that significantly increased (fold change > 2) from the zygote stage to the 8-cell stage in normal embryos, though not in arrested 8-cell embryos. **i** and **j** Gene ontology analysis of the transcripts of Group A **i** and Group B **j**.

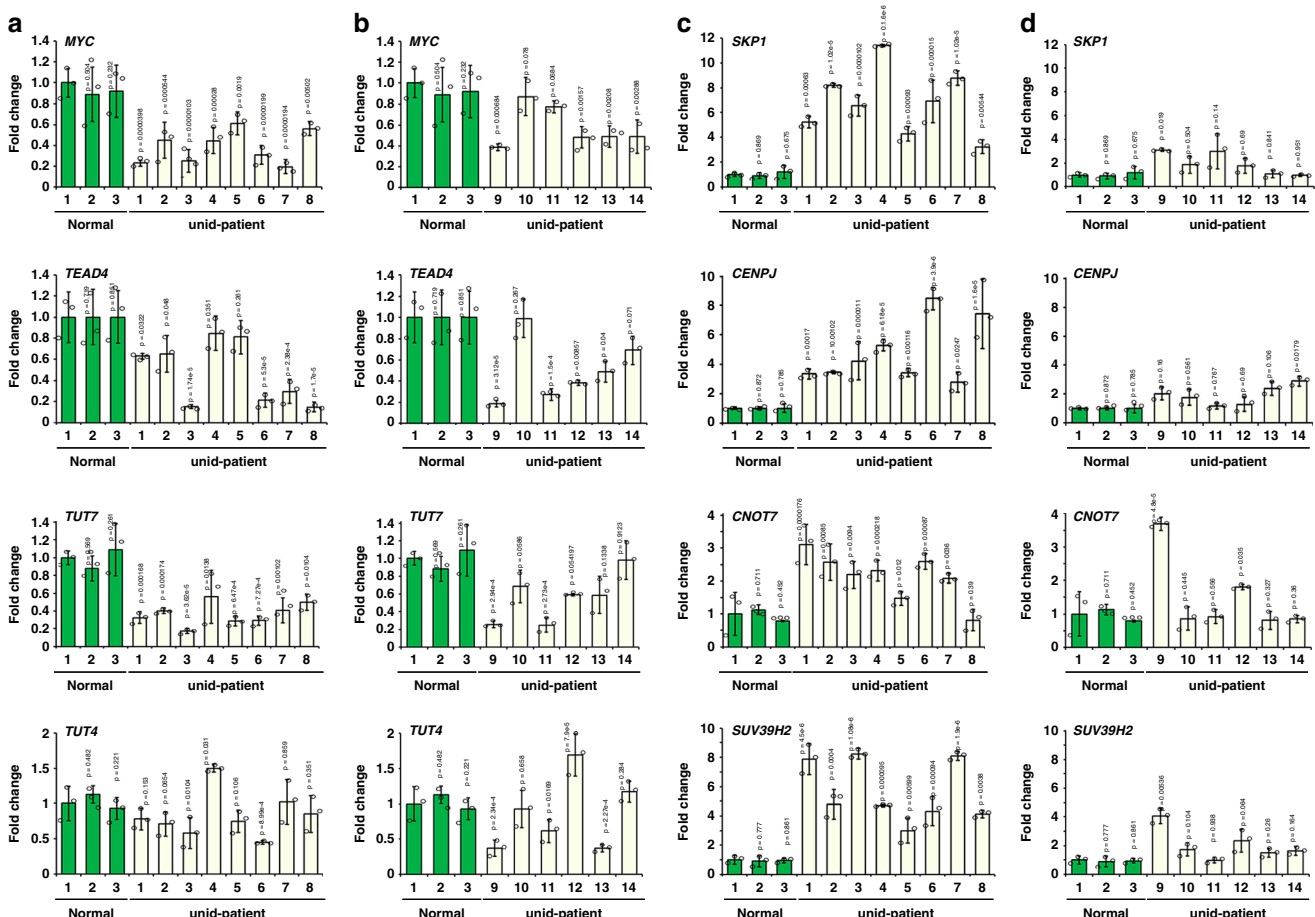

**Fig. 9 Levels of Z-decay-related transcripts in human embryos after IVF. a** and **b** The RT-qPCR results show the mRNA levels of key zygotic factors in normal and 8-cell stage-arrested embryos. 1–3, normal 8-cell embryos; 1–14, 8-cell stage-arrested embryos from different patients. **c** and **d** The RT-qPCR results show the mRNA levels of selected Z-decay transcripts in normal and 8-cell stage-arrested embryos. Data are presented as mean values ± SEM. P by one way ANOVA. n.s. non-significant. n = 3 independent experiments.

accumulated, leading to embryonic developmental retardation at the 8-cell stage.

It was unclear whether timely mRNA degradation occurs in oocytes that are arrested in meiosis I or II due to spindle assembly defects or whether M-decay depended on progression to meiosis II or even meiosis exit after fertilization[6]. In previous studies, we have performed experiments in mouse oocytes to address this unanswered question. We artificially arrested the maturing oocytes in meiosis I by treatment with nocodazole, which is a widely used microtubule disruptor. We then detected the degradation of selective mRNAs that should have been removed by M-

decay in these oocytes using RT-qPCR. The results showed that, while the degradation of these mRNAs was impaired by M-decay-associated genetic defects, they were not affected by nocodazole treatment[35]. In this study, we further provided in vivo evidence that M-decay in humans is not impaired by meiosis defects caused by *TUBB8* mutations. This is evidence that the delayed mRNA decay observed in some arrested zygotes is primarily due to a lack of M-decay factors, rather than secondary consequences of cell cycle arrest.

In the Z-decay in *Drosophila* and zebrafish embryos, micro-RNAs play an important role. Smaug, which is a master MZT

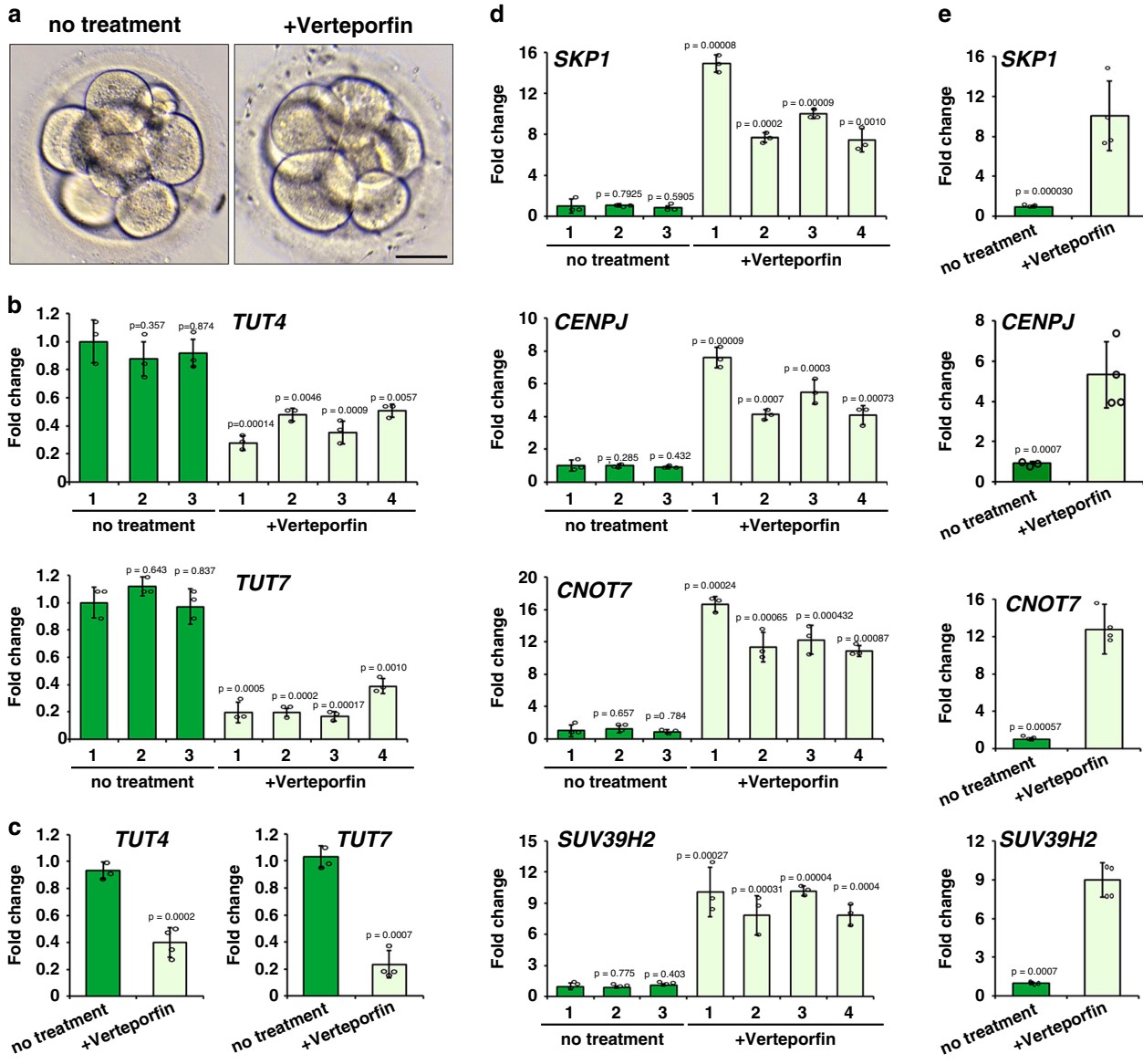

**Fig. 10 Effect of YAP inhibitor verteporfin on human early embryo development. a** Representative images of in vitro cultured human 8-cell embryos. Zygotes with 3PN were cultured with or without the presence of verteporfin and collected at the 8-cell stage. Scale bar = 50 μm. All embryos (n = 4 for each group) looked like this in three independently repeated experiments. **b** RT-qPCR results showing levels of *TUT4* and *TUT7* transcripts in the individual 8-cell embryos with or without verteporfin treatment. **c** One-way ANOVA test comparing *TUT4/7* differences at the 8-cell stage between verteporfin-treated and none-treated embryos. **d** RT-qPCR results showing levels of indicated transcripts in the individual 8-cell embryos with or without verteporfin treatment. **e** One-way ANOVA test comparing differences of transcript levels at the 8-cell stage between verteporfin-treated and none-treated embryos. In **b**–**e**, n = 3 independent experiments. Data are presented as mean values ± SEM. *P* by one-way ANOVA.

regulator in *Drosophila*, is required for zygotic synthesis of the miR-309 family of microRNAs, which targets several hundred maternal transcripts for degradation during MZT[46,47]. Zebrafish miR-430 is expressed at the onset of zygotic transcription and facilitates the deadenylation and clearance of maternal mRNAs during early embryogenesis[48]. However, it has also been reported that microRNA function is globally suppressed in mouse oocytes and early embryos[49]. Oocyte-specific deletion of *Dgcr8*, which encodes a protein that is specifically required for microRNA processing, does not affect the MZT in mice[49]. Thus, mRNAs of certain early zygotic genes may be the major zygotic transcription products that regulate Z-decay in mammals.

On the contrary, the involvement of de novo proteins translated from early zygotic transcripts in Z-decay is not clearly described in all model systems. Our studies suggest that early

zygotic expression of *Tead4* and *Tut4/7* may be required for Z-decay in both mice and humans. In 4-cell embryos derived from oocyte-specific *Yap1* knockout mice, maternal transcripts were accumulated, particularly those that were destined to be removed by the Z-decay pathway[14,50]. Similarly, in the arrested human embryos, the decreased expression of these factors (*BTG4*, *CNOT7*, *TEAD4*, *TUT4/7*) was closely associated with maternal mRNA degradation defects, which suggests that the maternal mRNA clearing pathway is highly conserved in vertebrate species.

Transcriptome analyses indicated that the Z-decay process is largely completed by the 8-cell stage in human embryos[18,28]. Different from its involvement in mice, zygotic transcription plays a more important role in the Z-decay of human maternal transcripts, probably due to a longer duration from ZGA to the completion of Z-decay in humans compared to that in mice[51]. It

has been noted in clinically assisted reproduction practices that many poor-quality embryos or embryos derived from aged oocytes were arrested at the 8~16-cell stage[52–54]. Therefore, the maternal and zygotic components in the Z-decay pathway may be key factors that determine the quality and developmental potential of human embryos.

## Methods

**Human oocyte and early embryo collection**. All of the oocytes and embryo were obtained with signed informed consent by the donor couples. The ovaries were stimulated using GnRH analogs combined with recombinant follicle stimulating hormone (FSH). Oocytes were obtained through follicle puncture at 36 h after hCG administration. The donated oocytes were randomly picked. The cumulus cells around each oocyte were removed using hyaluronidase treatment. MII oocytes were acquired from in vitro maturation of the immature (GV/MI) oocyte.

To collect early embryos, in vitro fertilized eggs were cultured until the 8-cell stage using a G-1 (Vitrolife) human embryos culture medium. The G-2 (Vitrolife) medium was used to culture the 8-cell embryos to the blastocyst stage. When normal embryos developed to the 8-cell stage at day 3 after fertilization, the embryos that were arrested at the 1-cell stage were collected. 8-cell embryos of the control groups were from abnormal zygotes (3PN fertilization) and collected at day 3 after fertilization. The arrested 8-cell embryos that have no signs of degeneration were collected at day 5 after fertilization. None of the donated oocytes were fertilized for the purposes of this study.

In this study, 17 GV oocytes were acquired from 6 donors; 39 arrested zygotes (3PN included) were acquired from 30 donors; and 23 arrested 8-cell embryos were acquired from 30 donors. The experiments performed in this study were approved and guided by the ethical committee of Guangdong Second Provincial General Hospital (Research license YY-2018-009-01) and the Reproductive & Genetic Hospital of CITIC-XIANGYA (Research license LL-SC-2019-030).

**Animals**. All the used mouse strains were of a C57B6 background. Wild type C57BL6 mice were obtained from the Zhejiang Academy of Medical Science, China. The experimental protocols that involved mice were approved by the Zhejiang University Institutional Animal Care and Research Committee (Approval # ZJU20170014), and mouse care and use was performed in accordance with the relevant guidelines and regulations.

**Mouse oocyte culture**. Female mice (21–23 days old) were injected with 5 IU of PMSG and were humanely euthanized after 44 h. Oocytes at the GV stage were harvested in M2 medium (M7167; Sigma-Aldrich) and cultured in mini-drops of M16 medium (M7292; Sigma-Aldrich) that were covered with mineral oil (M5310; Sigma-Aldrich) at 37 °C in a 5% $CO_2$ atmosphere.

**Microinjection of mouse oocyte**. All injections were performed using an Eppendorf transferman NK2 micromanipulator. GV oocytes were incubated in M2 medium with 2 μM milrinone to inhibit spontaneous GVBD and microinjected as 5–10 pL samples per zygote. The concentration of all microinjected RNAs was then adjusted to 1000 ng/μl. After microinjection, oocytes were washed and cultured in M16 medium at 37 °C with 5% $CO_2$.

**Immunofluorescence**. Oocytes and embryos were fixed in 4% paraformaldehyde in phosphate-buffered saline (PBS) for 30 min and permeabilized in PBS containing 0.3% Triton X-100 for 30 min. After being blocked with 1% bovine serum albumin in PBS, the oocytes were incubated with primary antibodies for 1 h and sequentially labeled with Alexa Fluor Cy3-conjugated or 488-conjugated secondary antibodies and 4′,6-diamidino-2-phenylindole (DAPI) for 30 min. A confocal microscope was used to image oocytes.

**In vitro transcription and preparation of mRNAs for microinjection**. To prepare mRNAs for microinjection, expression vectors were linearized and subjected to phenol/chloroform extraction and ethanol precipitation. Linearized DNAs were in vitro transcribed using the SP6 message mMACHINE Kit (Life, AM1340). Transcribed mRNAs were then added to poly (A) tails (~200–250 bp) using the mMACHINE Kit (Life, AM1350), recovered by lithium chloride precipitation, cleaned by ethanol, and lastly resuspended in nuclease-free water.

**Single cell RNA-Seq library preparation**. To remove the zona pellucida, the embryos were exposed to acidic Tyrode's solution (pH 2.5, Sigma, Cat#T1788) for 3–5 s and then washed thoroughly in PBS containing 0.5% bovine serum albumin (BSA) (Sigma, Cat#A1933). Single cells were placed into individual tubes that contained 4 μl of lysis buffer (1.96 μl of nuclease-free water, 1 μl of 10 mM dNTP mix (NEB, Cat#N0447), 0.1 μl of 40 U/ml RNase-inhibitor (NEB, Cat#M0314L), 0.04 μl of 10% Triton X-100 (Sigma, Cat#T8787), and 1 μl of 10 mM modified oligo-dT primer (5′-AACGCAGAGTACT30VN-3′)). After 3 min of cell lysis at 72 °C, Smart-seq2 reverse transcription reactions were performed. After the first-

strand reaction, the cDNA was amplified using a limited number of cycles (~13 cycles). Sequencing libraries were constructed from 500 pg of amplified cDNA using the TruePrep DNA Library Prep Kit V2 for Illumina (Vazyme, TD503) according to the manufacturer's instructions. Barcoded libraries were pooled and sequenced on the Illumina HiSeq X Ten platform in the 150 bp paired-end mode.

**RNA seq data analysis**. All the raw reads were first preprocessed using Trimmomatic (v0.35)[55] to remove sequencing adapters, trim low-quality bases from both read ends (with the parameters LEADING:3 TRAILING:3 SLIDINGWINDOW:4:15), and remove reads <36 bp in length. The clean reads were then mapped to the human reference genome of GRCh38 (without masking repeats) using STAR aligner (v2.5.2b)[56]. Ensemble genes were calculated using HTSeq (v0.6.1p1)[57]. The expression levels of each gene were quantified using normalized FPKM. Two-tailed Student's $t$-test was used to determine statistical significance of differences between samples. PCA clustering for different embryos was performed using the R prcomp function. Summaries of the RNA-seq data generated in this study are shown in Supplementary Tables 5 and 6.

**RNA isolation and real-time RT-PCR**. Oocytes or embryos were collected and lysed in 2 μl of lysis buffer (0.2% Triton X-100 and 4 IU RNase inhibitor) followed by reverse transcription with primer transcript II reverse transcriptase (Takara), according to the manufacturer's instructions. A real-time RT-PCR analysis was performed using the Power SYBR Green PCR Master Mix (Applied Biosystems, Life technologies) and an Applied Biosystems 7500 Real-Time PCR System. The respective cycle threshold (Ct) values were obtained, and relative mRNA levels were calculated by normalization to the endogenous *Gapdh* mRNA levels (internal control) using Microsoft EXCEL®. The gene expression levels were calculated by $2^{\Delta Ct}$ ($2^{\Delta Ct}$ (*genes−Gapdh*)). The relative transcript levels of the samples were compared to those of controls, and fold changes were determined. For each experiment, qPCR was performed in triplicate. Primer sequences are listed in Supplementary Table 7.

**Maternal transcript clustering**. The data was extracted from previously published dataset (GV data from GSE107746 and others from GSE36552; The datasets of α-amanitin treatment are from GSE101571). Maternal mRNAs with reliable sequence annotations and FPKM of >2 at the GV stage were retained for further analysis. Expression levels of each gene were added to one and then transformed by log2 in the following analysis. Cluster I–IV consisted of genes that satisfy the following formulas:

Cluster I: Expression (GV) > Expression (zygote) + 1; Expression (zygote) ≤ Expression (8-cell)+1.

Cluster II: Expression (GV) ≤ Expression (zygote)+1; Expression (GV) > Expression (zygote)−1; Expression (zygote)>Expression (8-cell)+1.

Cluster III: Expression (GV) > Expression (zygote)+1; Expression (zygote) > Expression (8-cell)+1.

Cluster IV: Expression (GV) ≤ Expression (zygote)+1; Expression (GV) > Expression (zygote)−1; Expression (zygote) ≤ Expression (8-cell)+1; Expression (zygote)>Expression (8-cell)−1.

**3′-UTR analysis**. The 3′-UTR sequences of humans (grch37) were extracted from the UCSC Table Browser. The conserved sequences 5′-UUUUAU/UUUUAAU-3′ and 5′-AAUAAA/AUUAAA-3′ were used to identify CPEs and PASs, respectively. The lengths of 3′-UTRs and numbers of CPEs and PASs in 3′-UTRs were calculated using an in-house Python script.

**Statistical analysis**. Results were presented as mean ± SEM. Most experiments included at least three samples and were repeated at least three times. The results for the two experimental groups were compared using two-tailed unpaired Student's $t$-tests and one way ANOVA. Values were considered statistically significant at $P < 0.05$, $P < 0.01$, and $P < 0.001$.

**Reporting summary**. Further information on research design is available in the Nature Research Reporting Summary linked to this article.

## Data availability

RNA-seq data have been deposited in the NCBI Gene Expression Omnibus database under accession code PRJNA603589. Source data are provided with this paper.

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

## Acknowledgements

This study is supported by: National Natural Science Foundation of China [91949108 to Q.-Q.S., 31930031, 31890781, 31671558 to H.-Y.F., 31801245 to S.L.]; Start-up Funding of Guangdong Second Provincial General Hospital [YY2019-001 to Q.-Q.S.]; National Key Research and Developmental Program of China [2017YFC1001500, 2016YFC1000600 to H.-Y.F., 2016YFC1000200 to G.L.]; Key Research and Development Program of Zhejiang Province [2017C03022 to H.-Y.F].

## Author contributions

H.-Y.F., X.-H.O., and G.L. conceived the project. H.-Y.F., Q.-Q.S., W.Z., X.-H.O., and G.L. designed and analyzed experiments. Q.-Q.S., S.L., Y.-W.W., S.Z., and L.G. performed experiments. X.-H.O., W.Z., and G.L. provided key reagents and materials. Q.-Q.S. and H.-Y.F. wrote the paper.

## Competing interests

The authors declare no competing interests.
