## [Peer Review File · Nature Communications]

Reviewers' Comments:

Reviewer #1:

Remarks to the Author:

I have been asked to conduct a bioethics review of "Dynamics and Clinical Relevance of Maternal mRNA Clearance during the Oocyte-to-embryo Transition in Humans." For the purposes of this study, the research team procured unfertilized human eggs and human embryos arrested at the zygote and 8-cell stages.

I examined the informed consent forms and the signed institutional approval forms for this complex study from the Reproductive Medicine Center of Guangdong Second Provincial General Hospital and from the Reproductive and Genetic Hospital of CITIC-XIANGYA. These procedures seem to be in order and seem to follow all applicable research guidelines and laws.

The team should make clearer that none of the donated oocytes were fertilized for the purposes of this study; rather, all embryos used by the study team were donated by couples seeking fertility treatment with informed consent and were sub-clinical in quality.

Reviewer #2:

Remarks to the Author:

The mechanism and functional importance of maternal mRNA decay during the human maternal-to-zygotic transition (MZT) have not yet been analyzed. This study describes the maternal (M)- and zygotic (Z)- mRNA decay dynamics before and after zygotic genome activation (ZGA). Moreover, the authors propose potential correlations of M-decay defects with zygotic arrest, and Z-decay defects with 8-cell arrest in arrested embryos of patients.

Overall, the analyses are convincing and the paper is well-written. However, the results are only correlative, and there is no mechanistic data on human embryos that support the authors' conclusion that an early developmental arrest of human embryos is linked to M- or Z-decay defects. Mechanistic data that demonstrate a function for factors proposed to be involved in M- or Z-decay in humans is also missing. Moreover, the mechanism of Z-decay in the mouse was characterized in reference 30 already, limiting the novelty of the study.

Main points:

1/ In the abstract, the authors state that "YAP1-TEAD4 transcription activators, TUT4/7-mediated mRNA 3'-oligouridylation, and BTG4/CCR4-NOT-induced mRNA deadenylation may all be involved in the regulation of maternal mRNA stability during human MZT". This conclusion was made based on the results of Fig. 2. However, in Fig. 2, the persistence of mRNAs encoding BTG4, CNOT7 and CNOT6L is not sufficient to suggest that they function in both M-decay and Z-decay. Knock-down experiments are required to draw this conclusion. In addition, the authors' conclusion could be strengthened if they test if protein levels of BTG4, CNOT7 and CNOT6L are changed during embryogenesis.

Similarly, describing the changes in the levels of mRNAs encoding TEAD4, TUT4 and TUT7 is not sufficient to conclude they are involved in Z-decay (i.e. the authors write "The results suggest that the Z-decay mechanism of YAP1/TEAD4-mediated zygotic Tut4/7 expression was conserved in human embryos). Also here, knock-down experiments are required to make this point.

2/ The authors argue that the developmental arrest of human embryos in the zygote and 8-cell stages is linked to defects in M-decay and Z-decay (i.e. in the abstract: "Defects of M-decay and Z-decay were detected with high incidence in embryos that were arrested at the zygote and 8-cell stages, respectively. These results suggest that M-decay and Z-decay pathway activities are required for effective human MZT, and thus contribute to the developmental potential of human

preimplantation embryos".) This conclusion is mainly based on the results in Fig. 4 to Fig. 7. The results of Fig. 4 and Fig. 5 indicate that M-decay was frequently impaired in arrested human zygotes, but this is not sufficient to support the authors' conclusion that M-decay was critical for the first embryonic cleavage after fertilization. Developmental arrest and defective M-decay may both be caused by another defect. It is hence not valid to conclude that the developmental arrest was caused by M-decay defects if there are no knock-down results. Similarly, the authors do not provide evidence that Z-decay defects cause 8-cell stage arrests. Although the results in Fig. 7 suggest some correlations between Z-decay defects and 8-cell stage arrests, these correlations are not sufficient to conclude that "Z-decay defects contribute to early developmental arrest of in vitro fertilized human embryos".

3/ In Fig. 5C and D, the BTG4 level of patient 10 and patient 15 was as low as that in patient 2-8, but mRNAs encoding EIF3K, ZP2, Mos and ZAR2 did not accumulate in these two patients as in patient 2-8. How do the authors explain that BTG4 reduction in patient 10 and 15 does not affect decay of EIF3K, ZP2, Mos and ZAR2 (Fig. 5B)? This seems to contradict the hypothesis that BTG4 is required for M-decay in humans, as in mouse. Is it possible that the developmental arrest and transcription changes are both caused by another defect?

4/ In Fig. 6G, "among the 4,074 Z-decay transcripts that were detected in normal embryos, 775 transcripts were stabilized in the arrested embryos". "Among these transcripts, 223 belonged to the previously identified ZGA-dependent Z-decay transcripts". This is a very small fraction. If the authors cannot provide additional evidence, it is not valid to conclude that "Human embryos with ZGA defects failed to remove maternal transcripts through Z-decay".

5/ Maternal mRNAs were classed into 4 clusters according to the level change from GV to 8-cell stage. Comparisons of these 4 groups between mouse and human would be helpful since they may utilize the same mechanisms for maternal mRNA decay.

6/ α -amanitin treatment has much stronger effects in humans than in the mouse (Fig. 1C and ref. 30). Is this related to the embryo stage (8-cell in human versus 2-cell in mouse)?

7/ According to Fig. 1C, it seems that some mRNAs are increased from zygote to 8-cell stage after α -amanitin treatment (Cluster III, inhibited). How do the authors explain this?

8/ According to Fig. 3D and the descriptions in methods "Human oocyte and early embryo collection", unidentified patients also have normal 8-cell embryos or blastocysts. Were M-decay or Z-decay affected in these embryos? These normal embryos can also be used as controls.

Minor points:

1/ Fig. S1A is important, and may need to be brought into the main figures.

2/ Why is TLE6 not included in Fig. 3C, as in Fig. 2A or Fig. 3F?

3/ In Fig. 6H, it is better to use names other than cluster I and cluster II, since they have been used to define 4 groups of maternal mRNAs.

4/ ANOVA test with post hoc analysis is a better choice to compare differences between controls and patients. It can reflect general differences between these two groups.

Reviewer #3:

Remarks to the Author:

Sha et al., 2020 NComms review

Sha et al., have analysed and performed transcriptomic profiling of human oocytes and normal and developmentally arrested embryos, to try and understand the importance of MZT in human and

whether defects in M-decay or Z-decay, two different stages of mRNA clearance during the MZT, could be linked to developmental arrest. We do not fully understand the importance or regulation of ZGA in mouse, and even less so in human, so this topic of this study is important. The authors define transcripts degraded in M or Z decay from previous RNA-seq data and ask whether there is evidence of defective decay in arrested zygotes or 8-cell human embryos. While the authors do find evidence of impaired decay in both cases they in many places claim a causative link between defective MZT and arrested development. Unfortunately, their data are not able to support more than an association between these two phenomena. Thus, I believe either the manuscript should be heavily toned down to only point to suggestive association between mRNA decay defects and embryo arrest in development, or several experiments would be needed to somehow provide mechanistic evidence to link the two. Further specific comments below

1) The authors affirm that ZGA in humans occurs at 4-8cell stage. However to define ZGA-dependent mRNA clearance the latest embryo timepoint is 8cell itself. It seems likely that many ZGA transcripts might only be significantly degraded >8cell. It would be good to confirm their ZGA-dependent degraded transcripts using extra embryo samples >8-cell - could be done by analysing other human embryo datasets.

2) Several cases of overstatement throughout "This observation suggests that BTG4 and CCR4-NOT may play important roles in human maternal mRNA decay, including both M-decay and Z-decay"
- just because factors are not degraded until 4-8 cell stage at mRNA level it doesn't mean that they are important for Z-decay.

Ditto for TUT7 expression (lines 172-173) - just because a factor is expressed in the correct window for Z-decay, it does not mean that it has a conserved role in Z-decay.

3) Data for 3F - suggests that M-decay happens as normal in TUBB8 mutant zygotes. However, we have no normal zygotes to compare to. 3C compares GV to MII oocytes, and we have no data here to show how much further levels would decline in normal zygotes. Thus although there are significant reductions in the indicated transcripts, it is not possible to know for sure whether there are still M-decay defects that have prevented a total/further degradation of transcripts

4) 241; "Overall, these results indicate that M-decay was frequently impaired in arrested 242 human zygotes and was critical for the first embryonic cleavage after fertilization

I appreciate the difficulty in working with rare and precious human samples, and potentially the need to sub-select sets of embryos based on correlation, and not as would be ideal - on genotype/mutation. However again, it is not possible to say more than correlation from these data and the above claims are overstated. The data implies that M-decay defects are correlated/associated with embryo arrest, but cannot say anything about causation. The authors could look at whether the subset of M-decay genes that are upregulated have any special mRNA features (binding sites etc) that might give a clue as to mechanism, but again without further experimental work it wouldn't be possible to say that M-decay is causative of an arrest.

5) Regarding the link between lower BTG4/CNOT expression - it would be helpful to show the data also as correlation graphs of expression which might make the authors' point more clear. I agree that there is an intriguing link between lower levels of these factors and retained M-decay defects. However, again it is unfortunately not possible by these data alone to claim a causative link.

These and other claims such as "338: Collectively, results of this study suggest that the defective maternal mRNA degradation machinery is a key contributing factor for the developmental failure of in vitro fertilized human embryos." are overstated.

6) In many cases multiple t-tests are used to compare levels expression of embryo samples to a

single control. These tests are not multiple-testing corrected, so overinflates significance. One-way ANOVA is more appropriate for comparing multiple samples to a single control. Authors should check all statistical testing such as above and adjust significance and conclusions accordingly, where relevant.

Re: NCOMMS-20-06223

Responses to Reviewers' comments:

Reviewer #1 (Remarks to the Author):

I have been asked to conduct a bioethics review of "Dynamics and Clinical Relevance of Maternal mRNA Clearance during the Oocyte-to-embryo Transition in Humans." For the purposes of this study, the research team procured unfertilized human eggs and human embryos arrested at the zygote and 8-cell stages.

I examined the informed consent forms and the signed institutional approval forms for this complex study from the Reproductive Medicine Center of Guangdong Second Provincial General Hospital and from the Reproductive and Genetic Hospital of CITIC-XIANGYA. These procedures seem to be in order and seem to follow all applicable research guidelines and laws.

The team should make clearer that none of the donated oocytes were fertilized for the purposes of this study; rather, all embryos used by the study team were donated by couples seeking fertility treatment with informed consent and were sub-clinical in quality.

Response: We appreciate the reviewer's careful examination of the bioethics and have clarified this point in the revised Materials and Methods.

--

Reviewer #2 (Remarks to the Author):

The mechanism and functional importance of maternal mRNA decay during the human maternal-to-zygotic transition (MZT) have not yet been analyzed. This study describes the maternal (M)- and zygotic (Z)- mRNA decay dynamics before and after zygotic genome activation (ZGA). Moreover, the authors propose potential correlations of M-decay defects with zygotic arrest, and Z-decay defects with 8-cell arrest in arrested embryos of patients.

Overall, the analyses are convincing and the paper is well-written. However, the results are only correlative, and there is no mechanistic data on human embryos that support the authors' conclusion that an early developmental arrest of human embryos is linked to M- or Z-decay defects. Mechanistic data that demonstrate a function for factors proposed to be involved in M- or Z-decay in humans is also missing. Moreover, the mechanism of Z-decay in the mouse was characterized in reference 30 already, limiting the novelty of the study.

Response: We greatly appreciate the reviewer's recognition of our manuscript as analytically convincing and well written. We agree with the reviewer that the mechanism of Z-decay in mouse was characterized in our recent paper (reference 30) and was described in the Introduction and Discussion. However, we politely argue that the current study of human MZT provides sufficient novel findings that, in our opinion, would be of great interest to the readers of Nature Communications. Specific reasons, some of which have already been included in the submitted manuscript, are as follows:

- 1) Unlike mouse embryos, in which the major wave of ZGA is initiated at the 2-cell stage, human embryos undergo the major wave of ZGA at the 8-cell stage. The proportion of human maternal transcripts that undergo ZGA-dependent clearance remains undetermined.
- 2) TUBB8 variants are genetic determinants of human oocyte maturation arrest that cause variable and mixed phenotypes in oocyte maturation and early embryo development. However, it is unclear if

the process of oocyte maturation-associated maternal mRNA decay was also disturbed in these mutant zygotes.

3) This the first study describing the M- and Z-decay of human maternal transcripts. The association of maternal mRNA decay with human early embryo arrest in assisted reproduction has not been reported previously.

Main points:

1/ In the abstract, the authors state that "YAP1-TEAD4 transcription activators, TUT4/7-mediated mRNA 3'-oligouridylation, and BTG4/CCR4-NOT-induced mRNA deadenylation may all be involved in the regulation of maternal mRNA stability during human MZT". This conclusion was made based on the results of Fig. 2. However, in Fig. 2, the persistence of mRNAs encoding BTG4, CNOT7 and CNOT6L is not sufficient to suggest that they function in both M-decay and Z-decay. Knock-down experiments are required to draw this conclusion. In addition, the authors' conclusion could be strengthened if they test if protein levels of BTG4, CNOT7 and CNOT6L are changed during embryogenesis.

Similarly, describing the changes in the levels of mRNAs encoding TEAD4, TUT4 and TUT7 is not sufficient to conclude they are involved in Z-decay (i.e. the authors write "The results suggest that the Z-decay mechanism of YAP1/TEAD4-mediated zygotic Tut4/7 expression was conserved in human embryos). Also here, knock-down experiments are required to make this point.

Response:

1) We agree with the reviewer that the current data provided correlative rather than causal evidence. In the revised manuscript, we have carefully toned down the wording and only speculated that these factors "**MAY BE** involved in the regulation of maternal mRNA stability during human MZT".

2) Meanwhile, in another study in which we collaborated with other groups, we have identified infertile women carrying *BTG4* mutations. The zygotes from these women were arrested at the 1-cell stage and exhibited defects in maternal mRNA degradation. The phenotypes were similar to those we have observed in *Btg4* knockout mice. This manuscript is currently published online by another journal (reference 41 of the revised manuscript). The identification of *BTG4* mutations in infertile women supports our hypothesis that BTG4/CCR4-NOT-induced mRNA deadenylation may be involved in the regulation of maternal mRNA stability during human MZT. Therefore, we described these results and indicated their connections with the current study in the revised Discussion.

3) Indeed, it would be desirable to have knock-down experiments to further elucidate the function of BTG4, CCR4-NOT, and TUT4/7 in M- and Z-decay. These experiments have been performed in mouse and the data suggested the indispensable role of these factors in MZT. However, in humans, only immature GV oocytes after superovulation and abnormal zygotes with 3PN after fertilization can be used for research purposes. These samples have very low developmental potential in culture and are therefore unsuitable for RNAi experiments. The bioethical regulations of our institution do not permit the use of healthy zygotes for this research purpose. Instead, we treated 3PN human zygotes with a YAP inhibitor (verteporfin) and found that it effectively repressed zygotic *TUT4/7* expression (revised Fig. 10). Consistent with our working model, Z-decay of maternal transcripts was impaired when these zygotes developed to the 8-cell stage. These new results provide more direct evidence that YAP and TUT4/7 are likely regulating maternal mRNA stability during human MZT.

4) It will certainly be interesting to determine if the protein levels of BTG4, CNOT7, and CNOT6L change during embryogenesis. However, at least 100 mouse oocytes are required to detect these proteins by western blot. It is practically impossible for us to collect enough human oocytes and embryos to

perform this experiment. Therefore, we performed immunofluorescence of BTG4 and CNOT7 in human GV oocytes and zygotes. The results in revised Fig. 3 revealed that these proteins were expressed in zygotes but were not detected in GV oocytes. This expression pattern is similar to that reported in mouse (Yu C. et al., Nature Structural & Molecular Biology 2016).

2/ The authors argue that the developmental arrest of human embryos in the zygote and 8-cell stages is linked to defects in M-decay and Z-decay (i.e. in the abstract: “Defects of M-decay and Z-decay were detected with high incidence in embryos that were arrested at the zygote and 8-cell stages, respectively. These results suggest that M-decay and Z-decay pathway activities are required for effective human MZT, and thus contribute to the developmental potential of human preimplantation embryos”.) This conclusion is mainly based on the results in Fig. 4 to Fig. 7.

The results of Fig. 4 and Fig. 5 indicate that M-decay was frequently impaired in arrested human zygotes, but this is not sufficient to support the authors’ conclusion that M-decay was critical for the first embryonic cleavage after fertilization. Developmental arrest and defective M-decay may both be caused by another defect. It is hence not valid to conclude that the developmental arrest was caused by M-decay defects if there are no knock-down results.

Similarly, the authors do not provide evidence that Z-decay defects cause 8-cell stage arrests. Although the results in Fig. 7 suggest some correlations between Z-decay defects and 8-cell stage arrests, these correlations are not sufficient to conclude that “Z-decay defects contribute to early developmental arrest of in vitro fertilized human embryos”.

Response:

1) As we have explained to the reviewer, homologous *BTG4* mutations caused maternal mRNA stabilization and zygotic arrest in human (a separate manuscript). In this study, we further observed a correlation between defects in maternal mRNA decay and zygotic arrest in human patients. Taken together, these results support our conclusion that M-decay is critical for the first embryonic cleavage after fertilization. We have described the identification of *BTG4* mutations and the impact of these findings on the current study in the revised Discussion.

2) We agree with the reviewer that the current results did not provide direct evidence that defects in Z-decay cause arrest at the 8-cell stage. In the revised manuscript, we carefully toned down our wording and only speculated that “Considering these correlations, Z-decay defects MAY contribute to early developmental arrest of in vitro fertilized human embryos”.

3/ In Fig. 5C and D, the BTG4 level of patient 10 and patient 15 was as low as that in patient 2-8, but mRNAs encoding EIF3K, ZP2, Mos and ZAR2 did not accumulate in these two patients as in patient 2-8. How do the authors explain that BTG4 reduction in patient 10 and 15 does not affect decay of EIF3K, ZP2, Mos and ZAR2 (Fig. 5B)? This seems to contradict the hypothesis that BTG4 is required for M-decay in humans, as in mouse. Is it possible that the developmental arrest and transcription changes are both caused by another defect?

Response:

1) We appreciate the reviewer’s concern. Although BTG4 is required for M-decay in mouse, not all maternal transcripts are degraded through a BTG4-dependent pathway. Particularly, CNOT6L and its RNA-binding adaptor ZFP36L2 mediate the early degradation of unstable maternal mRNAs at the onset of meiotic maturation (Sha QQ et al, EMBO Journal 2018). As shown in Fig. 5B of this paper, *Eif3k* is degraded in a CNOT6L-dependent but BTG4-independent manner. RNA-seq indicated that other

transcripts stabilized and accumulated more than twice as much in CNOT6L-deletion oocytes compared to WT oocytes during the GV to MII transition (Expanded Table 1). This observation can explain why *BTG4* reduction in patients 10 and 15 does not affect the decay of *EIF3K*, *MOS*, and *ZAR2*: some transcripts are degraded by a CNOT6L-dependent pathway despite the low expression of *BTG4*.

Gene name	FPKM						
	WT(GV)	WT(MII)	Cnot6l ^{-/-} (GV)	Cnot6l ^{-/-} (MII)	WT(MII/GV)	Cnot6l ^{-/-} (MII/GV)	[Cnot6l ^{-/-} (MII/GV)]/[WT(MII/GV)]
Eif3k	392.73	29.19	286.17	97.25	0.07	0.34	4.57
Zp2	1708.16	145.51	1235.51	678.35	0.09	0.55	6.45
Mos	714.27	215.75	591.19	362.06	0.3	0.61	2.03
Zar1l (Zar2)	558.57	1.79	547.11	49.73	0	0.09	28.36

Expanded Table 1: FPKM of transcripts in GV and MII oocytes derived from WT and *Cnot6l*^{-/-} mice (Sha QQ et al, EMBO Journal 2018).

2) Furthermore, human samples obtained from the clinic often vary significantly across many factors, including patient age, genetic background, living environment, diet, and other factors. These factors cannot be strictly controlled as they are in experiments using a mouse model. Therefore, unexplained variations that do not conform to major claims, such as patient 15 in Fig. 7, are occasionally observed. With these considerations in mind, we hope that the reviewers can understand.

4/ In Fig. 6G, “among the 4,074 Z-decay transcripts that were detected in normal embryos, 775 transcripts were stabilized in the arrested embryos”. “Among these transcripts, 223 belonged to the previously identified ZGA-dependent Z-decay transcripts”. This is a very small fraction. If the authors cannot provide additional evidence, it is not valid to conclude that “Human embryos with ZGA defects failed to remove maternal transcripts through Z-decay”.

Response:

1) We did not perform the α -amanitin experiment ourselves; the data were extracted from a previously published dataset (Yan et al., Nat Struct Mol Biol, 2013; Chen et al., Nature, 2019) and analyzed. The absolute FPKMs can vary among different datasets due to differences in input RNA quantity, the efficiency of reverse transcription, and detection sensitivity. Therefore, it is common to observe fewer overlaps of transcriptomic datasets published by different groups.

2) To further demonstrate this fact, we compared the transcripts within a single category (degraded from the zygote to 8-cell stage) from three different datasets. This revealed that the numbers of degraded genes were comparable (4047, 3368, 3114), but only 1,145 genes ($1,145/4,047 = 28.29\%$) were shared among these three different datasets (Expanded Fig. 1). Therefore, it is conceivable that there will be even fewer overlapping genes when comparing among multiple categories, such as in Fig. 8g.

3) Despite the technical issue explained above, we realized that the gene numbers are indeed too small, as the reviewer has indicated. As such, we made additional explanations and significantly toned down our language in the revised manuscript. However, we conclude that “Human embryos with ZGA defects failed to remove maternal transcripts through Z-decay” mainly based on the results in Fig. 1d that inhibition of ZGA by α -amanitin prevented the maternal mRNA clearance.

Expanded Figure 1: Overlap of human genes with transcripts degraded from the zygote to 8-cell stage in three different datasets.

5/ Maternal mRNAs were classed into 4 clusters according to the level change from GV to 8-cell stage. Comparisons of these 4 groups between mouse and human would be helpful since they may utilize the same mechanisms for maternal mRNA decay.

Response: We appreciate the reviewer's helpful suggestion and compared the four clusters of maternal transcripts between mouse and human. The results are shown in revised Fig. 2B. Despite these four clusters being defined by similar criteria, the genes in each cluster were significantly different in human and mouse. There may be two reasons causing these differences:

- 1) The mouse and human genes were extracted from different datasets generated by different groups. As we explained in our response to the reviewer's previous comment, the absolute FPKMs can vary among different datasets due to differences in input RNA quantity, the efficiency of reverse transcription, and detection sensitivity. Therefore, it is common to observe fewer overlaps between the human and mouse transcriptomes.
- 2) To demonstrate the differences between human and mouse maternal transcriptomes, we directly compared the transcripts in human and mouse GV oocytes in revised Fig. 2A. This revealed that only half of the transcriptomes were overlapping, indicating that the homology between human and mouse maternal transcriptomes is low.

Expanded Figure 2: Comparisons of maternal mRNA degradation pattern between mouse and human.

6/ α -amanitin treatment has much stronger effects in humans than in the mouse (Fig. 1C and ref. 30). Is this related to the embryo stage (8-cell in human versus 2-cell in mouse)?

Response: Yes, we think this is at least partially related to the embryo stage. Because the major wave of ZGA occurs at the 8-cell stage in human and the 2-cell stage in mouse, human maternal transcripts are degraded more slowly than mouse maternal transcripts; human zygotic factors have a longer time window to function and may play more important roles in mediating Z-decay than those in mouse. In addition, this observation may reflect an interspecies difference between human and mouse: Z-decay in human may depend more stringently on zygotic factors; however, maternal BTG4 and CCR4-NOT are also able to mediate the degradation of Z-decay transcripts, albeit at a slower rate (Fig. 4 of reference 30). As a result, inhibition of zygotic transcription by α -amanitin treatment has a much stronger effect in human than in mouse.

7/ According to Fig. 1C, it seems that some mRNAs are increased from zygote to 8-cell stage after α -amanitin treatment (Cluster III, inhibited). How do the authors explain this?

Response:

1) The data were extracted from a previously published dataset (Yan et al, Nat Struct Mol Biol, 2013; Chen et al., Nature, 2019) and analyzed. The expression of some mRNAs appears to have increased due to the differences in the datasets. The datasets of α -amanitin treatment were only published in Chen et al., Nature, 2019. However, this paper did not generate RNA-seq data in zygotes. Therefore, we had to perform our analysis using multiple different datasets. The absolute FPKMs can vary among different datasets due to differences in input RNA quantity and detection sensitivity.

2) To further address the reviewer's concern, we reanalyzed the transcriptomes at the GV, 2-cell (instead of zygote), and 8-cell stages, using data available from the same study (Chen et al., Nature, 2019). In our opinion, the transcriptomes of zygotes and 2-cell stage embryos are not dramatically different because the major wave of ZGA has not yet occurred. As shown in the attached figure, maternal transcript degradation is inhibited, but no transcript levels are increased from the 2-cell to 8-cell stage when the embryos were treated with α -amanitin beginning from the zygote stage (Expanded Fig. 3). This result further indicates that the mRNAs that show increases in revised Fig. 1D

are caused by the combined analyses of different datasets.

Expanded Figure 3: Dynamics of maternal mRNA clearance in human preimplantation embryos. **A:** Degradation patterns of human maternal transcripts at the GV, 2-cell, and 8-cell stages. Transcripts with FPKM > 2 at the GV stage were selected and further analyzed. Each light blue line represents the expression level of one gene, and the middle red line represents the median expression level of the cluster. **B:** Degradation patterns of maternal transcripts in human embryos with or without α -amanitin treatment. Transcripts with FPKM (8-cell)/FPKM (2-cell) < 0.5 were selected for analysis. Each light blue line represents the expression level of one gene. The middle red line represents the median expression level of the cluster. The green line represents the median expression level of the cluster after α -amanitin treatment.

8/ According to Fig. 3D and the descriptions in methods "Human oocyte and early embryo collection", unidentified patients also have normal 8-cell embryos or blastocysts. Were M-decay or Z-decay affected in these embryos? These normal embryos can also be used as controls.

Response: Indeed, it would be very desirable to use these normal embryos as controls. However, these embryos were used for uterine transplantation or were frozen for later transplantation (if the initial blastocyst transplantation failed to establish pregnancy). We did not receive permission from the patients to use these normal embryos for experimental purposes.

Minor points:

1/ Fig. S1A is important, and may need to be brought into the main figures.

Response: As the reviewer suggested, we have brought this result into the revised Fig. 3.

2/ Why is TLE6 not included in Fig. 3C, as in Fig. 2A or Fig. 3F?

Response: During normal MZT in mouse, *Tle6* mRNA remains stable during oocyte maturation and is not degraded until the zygote stage (Expanded Fig. 4). Revised Fig. 4C primarily shows the degraded mRNA at the MII stage. *TUBB8* mRNAs (WT or mutant) were microinjected at the GV stage and the *in vitro* matured oocytes were fertilized *in vitro*. Following these procedures, the fertilization rate was very low. As such, we did not show mRNA levels in zygotes derived from the microinjected oocytes.

Because *Tle6* was not yet degraded in WT or *TUBB8* mutant-expressing oocytes at the MII stage, it was not included in original Fig. 3C.

Expanded Figure 4: RT-PCR shows the mRNA level of *Tle6* in mouse oocytes.

3/ In Fig. 6H, it is better to use names other than cluster I and cluster II, since they have been used to define 4 groups of maternal mRNAs.

Response: We appreciate the reviewer's helpful suggestion and have corrected the names as Group A and Group B.

4/ ANOVA test with post hoc analysis is a better choice to compare differences between controls and patients. It can reflect general differences between these two groups.

Response: We appreciate the reviewer's helpful suggestion. We analyzed the differences between controls and patients and have added this result to the revised Table S2 and S4.

--

Reviewer #3 (Remarks to the Author):

Sha et al., have analysed and performed transcriptomic profiling of human oocytes and normal and developmentally arrested embryos, to try and understand the importance of MZT in human and whether defects in M-decay or Z-decay, two different stages of mRNA clearance during the MZT, could be linked to developmental arrest. We do not fully understand the importance or regulation of ZGA in mouse, and even less so in human, so this topic of this study is important. The authors define transcripts degraded in M or Z decay from previous RNA-seq data and ask whether there is evidence of defective decay in arrested zygotes or 8-cell human embryos. While the authors do find evidence of impaired decay in both cases they in many places claim a causative link between defective MZT and arrested development. Unfortunately, their data are not able to support more than an association between these two phenomena. Thus, I believe either the manuscript should be heavily toned down to only point to suggestive association between mRNA decay defects and embryo arrest in development, or several experiments would be needed to somehow provide mechanistic evidence to link the two. Further specific comments below

1) The authors affirm that ZGA in humans occurs at 4-8cell stage. However to define ZGA-dependent mRNA clearance the latest embryo timepoint is 8cell itself. It seems likely that many ZGA transcripts

might only be significantly degraded >8cell. It would be good to confirm their ZGA-dependent degraded transcripts using extra embryo samples >8-cell - could be done by analysing other human embryo datasets.

Response: We appreciate the reviewer's helpful suggestion. To assess whether all Cluster IV transcripts remained stable beyond the 8-cell stage or if a subset of transcripts were degraded after this timepoint, we also analyzed transcript levels at the morula stage (Expanded Fig. 5). This analysis indicated that only 176 of the 1531 Cluster IV transcripts were, in fact, degraded between the 8-cell and morula stage (Expanded Fig. 5A). The majority of Cluster IV transcripts remained stable between the 8-cell and morula stage (Expanded Fig. 5B). Therefore, ZGA-dependent mRNA clearance primarily occurs at the 8-cell stage. We have added this result to the revised Fig. 1.

Expanded Figure 5: Dynamics of human cluster IV maternal transcripts from the GV to morula stages. **A:** Transcripts with fold change (8-cell/morula) > 2. **B:** Transcripts with $1/2 < \text{Fold change (8-cell/morula)} < 2$. Each light blue line represents the expression level of one gene, and the middle red line represents the median expression level of the cluster.

2) Several cases of overstatement throughout “This observation suggests that BTG4 and CCR4-NOT may play important roles in human maternal mRNA decay, including both M-decay and Z-decay” – just because factors are not degraded until 4-8 cell stage at mRNA level it doesn't mean that they are important for Z-decay.

Ditto for TUT7 expression (lines 172-173) – just because a factor is expressed in the correct window for Z-decay, it does not mean that it has a conserved role in Z-decay.

Response:

1) We agree with the reviewer that the current BTG4 and TUT7 expression data in the correct window for Z-decay indicates a correlation, rather than providing causal evidence. In the revised manuscript, we carefully toned down our language and only speculated that these factors **MAY** have conserved roles in Z-decay.

2) Meanwhile, in another study in which we collaborated with other groups, we have identified infertile women carrying *BTG4* mutations. The zygotes from these women arrest at the 1-cell stage and exhibit defects in maternal mRNA degradation. The phenotypes were similar to those we have observed in *Btg4* knockout mice (reference 41 of the revised manuscript). The identification of *BTG4* mutations in infertile women supports our hypothesis that BTG4/CCR4-NOT-induced mRNA deadenylation may be involved in the regulation of maternal mRNA stability during human MZT. Therefore, we described these results and indicated their connections with the current study in the revised Discussion.

3) In addition, we treated the 3PN human zygotes with a YAP inhibitor (verteporfin) and found that it

effectively repressed zygotic TUT4/7 expression (revised Fig. 10). Consistent with our working model, Z-decay of maternal transcripts was impaired when these zygotes developed to the 8-cell stage. These new results provide more direct evidence that YAP and TUT4/7 are likely regulating maternal mRNA stability during human MZT.

4) As the reviewer suggested, we carefully avoided overstatement in the revised manuscript. We toned down the language and emphasized only the association between maternal mRNA decay and preimplantation embryo development.

3) Data for 3F – suggests that M-decay happens as normal in TUBB8 mutant zygotes. However, we have no normal zygotes to compare to. 3C compares GV to MII oocytes, and we have no data here to show how much further levels would decline in normal zygotes. Thus although there are significant reductions in the indicated transcripts, it is not possible to know for sure whether there are still M-decay defects that have prevented a total/further degradation of transcripts

Response: As the reviewer suggested, (1) we have added RT-qPCR results from normal mouse zygotes for comparison with those of GV and MII oocytes in revised Fig. 4C. (2) We also added RT-qPCR results from 3PN human zygotes for comparison with those of GV and MII oocytes in revised Fig. 3A. These results revealed that the levels of these M-decay transcripts were comparable in MII oocytes and in zygotes, suggesting that degradation of these transcripts was largely completed by the MII stage and there is no significant degradation during the MII-to-zygote transition, in both mouse and human. (3) RT-qPCR results showed that the indicated mRNAs in the arrested zygotes carrying maternal *TUBB8* mutations were at comparable levels with those in 3PN zygotes, suggesting that they are totally degraded, and not further degraded in maternal *TUBB8* mutated zygotes. (revised Fig. 5c and Fig. S1b).

4) 241; “Overall, these results indicate that M-decay was frequently impaired in arrested human zygotes and was critical for the first embryonic cleavage after fertilization

I appreciate the difficulty in working with rare and precious human samples, and potentially the need to sub-select sets of embryos based on correlation, and not as would be ideal – on genotype/mutation. However again, it is not possible to say more than correlation from these data and the above claims are overstated. The data implies that M-decay defects are correlated/associated with embryo arrest, but cannot say anything about causation. The authors could look at whether the subset of M-decay genes that are upregulated have any special mRNA features (binding sites etc) that might give a clue as to mechanism, but again without further experimental work it wouldn't be possible to say that M-decay is causative of an arrest.

Response:

1) We appreciate the reviewer's kind understanding. As mentioned above, we have recently observed that a homologous *BTG4* mutation caused maternal mRNA stabilization and zygotic arrest in human (reference 41 of the revised manuscript). In this study, we further observed a correlation between defects in maternal mRNA decay and zygotic arrest in human patients. Taken together, these results support our conclusion that M-decay is critical for the first embryonic cleavage after fertilization. We have described the identification of *BTG4* mutations and the impact of these findings to the current study in the revised Discussion.

2) Again, we agree with the reviewer that many results in this manuscript provided correlation rather than causal evidence. In the revised manuscript, we carefully toned down our language and only

speculated that “our data imply that M-decay defects are correlated/associated with embryo arrest”.

5) Regarding the link between lower BTG4/CNOT expression – it would be helpful to show the data also as correlation graphs of expression which might make the authors' point more clear. I agree that there is an intriguing link between lower levels of these factors and retained M-decay defects. However, again it is unfortunately not possible by these data alone to claim a causative link.

Response:

1) We appreciate the reviewer's helpful suggestion. We have attempted correlation analyses between BTG4/CNOT expression and M-decay transcripts but the results did not show significant correlations. This may be due to complementary functions of different M-decay factors; although BTG4 is required for M-decay in mouse, not all maternal transcripts are degraded through a BTG4-dependent pathway. Particularly, CNOT6L and its RNA-binding adaptor ZFP36L2 mediate the early degradation of unstable maternal mRNAs at the onset of meiotic maturation (Sha QQ et al, EMBO Journal 2018). For more detailed explanations, please see our responses to reviewer 2's comment 3.

2) Generally, hundreds of samples are required for reliable correlation analyses. However, we only have embryos from less than 20 patients to analyze. Therefore, significant variations among samples may mask the correlation between BTG4/CNOT6L expression and the levels of M-decay transcripts.

2) As explained above, the identification of BTG4 mutations in infertile human patients has indicated a causative link between M-decay defects and zygotic arrest in human. We have addressed this issue in the revised Discussion.

These and other claims such as “338: Collectively, results of this study suggest that the defective maternal mRNA degradation machinery is a key contributing factor for the developmental failure of in vitro fertilized human embryos.” are overstated.

Response: We reworded these sentences to avoid overstatement, as the reviewer suggested.

6) In many cases multiple t-tests are used to compare levels expression of embryo samples to a single control. These tests are not multiple-testing corrected, so overinflates significance. One-way ANOVA is more appropriate for comparing multiple samples to a single control. Authors should check all statistical testing such as above and adjust significance and conclusions accordingly, where relevant.

Response: We appreciate the reviewer's helpful suggestion. We reanalyzed statistical testing using a One-way ANOVA and adjusted significance in revised figure 7 and 9, Supplementary Table S1 and S4.

Reviewers' Comments:

Reviewer #1:

Remarks to the Author:

Thank you for your responses to my queries in my first review. I have no further concerns at this time.

Reviewer #2:

Remarks to the Author:

Most of the reviewer's comments were addressed in the revised manuscript. Reference 41 supports the hypothesis that BTG4/CCR4-NOT-induced mRNA deadenylation may be involved in the regulation of maternal mRNA stability during human MZT. Protein level changes of BTG4 and CNOT7 from GV to 8-cell embryos also support this hypothesis. The YAP inhibitor (verteporfin) experiment suggests involvement of YAP and TUT4/7 in Z-decay. These additional experiments make the conclusions more reliable. The toned-down phrasing is also more accurate than before. Additional points:

1) Datasets from different studies were used in this paper, which might be a cause of variations.

The authors need to clearly specify the datasets they used and outline the analysis methods in the "Materials and Methods" section.

2) Numbers of human oocytes used for each experiment (from each patient) need to be clarified.

Re: NCOMMS-20-06223A

Responses to Reviewers' comments:

Reviewer #1 (Remarks to the Author):

Thank you for your responses to my queries in my first review. I have no further concerns at this time.

Response: We thank the reviewer's careful examination of the bioethics of our manuscript.

--

Reviewer #2 (Remarks to the Author):

Most of the reviewer's comments were addressed in the revised manuscript. Reference 41 supports the hypothesis that BTG4/CCR4-NOT-induced mRNA deadenylation may be involved in the regulation of maternal mRNA stability during human MZT. Protein level changes of BTG4 and CNOT7 from GV to 8-cell embryos also support this hypothesis. The YAP inhibitor (verteporfin) experiment suggests involvement of YAP and TUT4/7 in Z-decay. These additional experiments make the conclusions more reliable. The toned-down phrasing is also more accurate than before.

Additional points:

1) Datasets from different studies were used in this paper, which might be a cause of variations. The authors need to clearly specify the datasets they used and outline the analysis methods in the "Materials and Methods" section.

Response: we specified the datasets used in this paper and outlined the analysis in Materials and Methods.

2) Numbers of human oocytes used for each experiment (from each patient) need to be clarified.

Response: we summarized the number of each cell type and donors used in this study in Materials and Methods.